# Beam-like models for the analyses of curved, twisted and tapered HAWT blades in large displacements

Giovanni Migliaccio[1], Giuseppe Ruta[2], Stefano Bennati[1], Riccardo Barsotti[1]

[1]Civil and Industrial Engineering, University of Pisa, Pisa, 56122, Italy
[2]Structural and Geotechnical Engineering, University "La Sapienza" of Roma, Roma, I-00184, Italy

*Correspondence to*: Giovanni Migliaccio (giovanni.migliaccio.it@gmail.com)

**Abstract.** The continuous effort to better predict the mechanical behaviour of complex beam-like structures like wind turbine blades is strictly related to requirements of performance improvement and costs reduction. But new design approaches and the increasing flexibility of those structures make their aero-elastic modelling ever more challenging. For the
structural part of this modelling, the best compromise between computational efficiency and accuracy can be obtained by a schematization based on suitable beam-like elements. This paper addresses the modelling of the mechanical behaviour of beam-like structures which are curved, twisted and tapered in their reference unstressed state, undergo large displacements, in- and out-of-plane cross-sectional warping and small strains. A suitable model for the problem at hand is proposed. Analytical and numerical results obtained by applying the proposed modelling approach, as well as comparison with 3D-
FEM results, are illustrated.

## 1 Introduction

In the process of improving horizontal axis wind turbines (HAWT) performance new methods are continuously being sought for capturing more energy, developing more reliable structures, and lowering the cost of energy (Wiser, 2016). Such goals can be achieved through the use of advanced materials, the optimization of the aerodynamic and structural behaviour of the
blades, and the exploitation of load control techniques (see, for example, Ashwill 2010, Bottasso 2012, Stäblein 2017). But new design strategies and the increasing flexibility of those structures make the modelling of their aero-elastic behaviour ever more challenging. For the structural part of this modelling, schematizing the blades through suitable beam-like elements can be the best compromise between computational efficiency and accuracy. But modern blades are complex beam-like structures. They can be curved, twisted and also tapered in their unstressed state. Even not considering the complexities
related to the materials and loading conditions, their shape alone is sufficient to make the mathematical description of their mechanical behaviour a very challenging task. This paper addresses the modelling of the mechanical behaviour of structures of this kind, with a particular focus on their main geometrical characteristics, such as the twist and taper of their cross-sections, the in- and out-of-plane warping of their cross-sections, and the large deflections of their reference centre-line.

Over the years several theories have been developed for beam-like structures (see, for example, Love 1944, Antmann 1966, and Rubin 1997). This is because beam models have historically been used in many fields, from helicopter rotor blades in aerospace engineering to bridges components in civil engineering and surgical tools in medicine. Nevertheless, the interest in advanced theories for complex beam-like structures has led to further researches also in recent years, due to the continuous need of ever more rigorous and application-oriented models. In this paper the attention is focused on the effects of important geometrical characteristics of those structures, such as the curvature of their centre-line, as well as the twist and the taper of their cross-sections. After an introduction to modelling approaches for structures of this kind (section 2), a suitable model for the problem at hand is proposed (section 3). Finally, analytical results and numerical examples obtained by applying the proposed modelling approach to reference beam-like structures are illustrated (sections 4 and 5).

## 2 Overview of modelling approaches

Aero-elastic modelling of modern blades can be addressed by means of different approaches (Wang 2016a). Those ones based on 3D FEM and beam models are two main choices for the structural part of this modelling. Although 3D FEM approaches can be very accurate and flexible, they can be computationally expensive for the analyses of complex systems, especially if CFD analyses are executed in parallel. The overall computational cost can be reduced if faster aerodynamic models are used, such as the blade element momentum (BEM) model, but even this solution may not be efficient enough for aero-elastic analyses and multi-objective optimization tasks. The coupling of BEM and suitable beam models can be the best compromise between computational efficiency and accuracy. In this work we focus the attention on mathematical models to simulate the mechanical behaviour of complex beam-like structures (hereafter referred to as beam-like models, or BLM), which can be curved, twisted and also tapered in their unstressed state, undergo large deflections, in- and out-of-plane cross-sectional warping and small strain. Suitable models are needed to simulate the behaviour of those structures. In general, classical beam models (see, for example, Love 1944), which include extension, twist and bending, as well as the Reissner's formulation (1981), also accounting for transverse shear deformation, may not be sufficient. Geometrically exact models are a better choice, but a way to put them into a suitable form for engineering applications is usually needed (Antman 1966). In general, suitable models should be both rigorous and application-oriented, two important requirements pursued over the years by many investigators (e.g. Giavotto 1983, Simo 1985, Ibrahimbegovic 1995, Ruta 2006, Pai 2011, and Yu 2012).

For over a century researchers have sought to represent beam-like structures by means of 1D models. Several theories have been developed, from the elementary Euler-Bernoulli theory, to the classical theory which includes Saint-Venant torsion, up to more refined theories, such as the Timoshenko theory for transverse shear deformations, the Vlasov theory for torsional warping restraint, and 3D beam theories which include 3D warping fields. Broadly speaking, beam theories can be grouped into engineering and mathematical theories. Several engineering theories are based on ad-hoc corrections to simpler theories (e.g. Rosen 1978), others are based on geometrically exact approaches (such as Wang 2016b and Hodges 2018). Among the mathematical theories, some approaches are based on the directed continuum (Rubin 2000), some others exploit asymptotic

methods (Yu 2012). The reason for such a large amount of works on beam theory is that it has always found application in many fields. By way of example, many approaches have been developed for helicopter rotor blades with an initial twist, but pre-twisted rods have always attracted the interest of many researchers in several fields (Rosen 1991). In the 1990's, Kunz (1994) provided an overview on modelling methods for rotating beams, illustrating how engineering theories for rotor blades evolved over the years. In those same years, Hodges (1990) reviewed the modelling approaches for composite rotor blades, discussing the importance of 3D warping and deformation coupling. More recently, Rafiee (2017) discussed vibrations control issues in rotating beams, summarizing beam theories and complicating effects, such as non-uniform cross-sections, initial curvatures, twist and sweep. It seems that, unlike the case of the pre-twisted rods, published results for curved rotating beams with initial taper and sweep are quite scarce, although all these geometrical characteristics can play an important role.

Up to now much has been done to develop powerful beam theories. However, there is still a gap between existing theories and those that could be suitable for complex beam-like structures. In general, the geometry of the reference and current states must be appropriately described. The curvature, twist and taper are important design features and should be explicitly included in the model. The analysis should not be restricted to small displacements. The model should provide the strain and stress fields in the three-dimensional beam-like structure, be rigorous and usable by engineers, and provide classical results when applied to prismatic isotropic homogeneous beams. Following these guidelines, a mathematical model to simulate the mechanical behaviour of the considered beam-like structures is proposed hereafter.

## 3 Mechanical model for complex beam-like structures

Here we are concerned with developing a mathematical model to describe the mechanical behaviour of beam-like structures which are curved, twisted and tapered in their reference state and undergo large displacements. One of the main issues with such a task is how to describe the motion of the structure (see, for example, Simo 1985, Ruta 2006, Pai 2014). The approach considered in this work is to imagine a beam-like structure as a collection of plane figures (i.e. the cross-sections) along a regular and simple three-dimensional curve (i.e. the centre-line). We assume that each point of each cross-section in the reference state moves to a position in the current state through a global rigid motion on which a local general motion is superimposed. In this manner, the cross-sectional deformation can be examined independently of the global motion of the centre-line. So, it is possible to consider the global motion to be large, while the local motion and the strain may be small.

### 3.1 Kinematics and strain measures

We begin by introducing two local triads of orthogonal unit vectors. The first one is the local triad, $b_i$, in the reference state, with $b_1$ aligned to the tangent vector of the reference centre-line. This frame is a function of the reference arch-length s, that is $b_i = b_i(s)$. The second local triad, $a_i$, is a suitable image of the local triad $b_i$ in the current state. This frame is a function of the reference arch-length s and the time t, that is $a_i = a_i(s,t)$. In general, $a_1$ is not required to be aligned to the tangent vector of the current centre-line. See Figure 1. It shows a schematic representation of the reference (left) and current (right) states of a

beam-like structure. The generic cross-section in the reference state is contained in the plane of the vectors $b_2$ and $b_3$. In the current state, if the cross-section remains plane (i.e. un-warped), it can belong to the plane of the vectors $a_2$ and $a_3$. But the generic cross-section may not remain plane. So, we consider that its current (warped) state is reached by superimposing an additional motion to the positions of the points of the un-warped cross-section, as in Figure 1 (right).

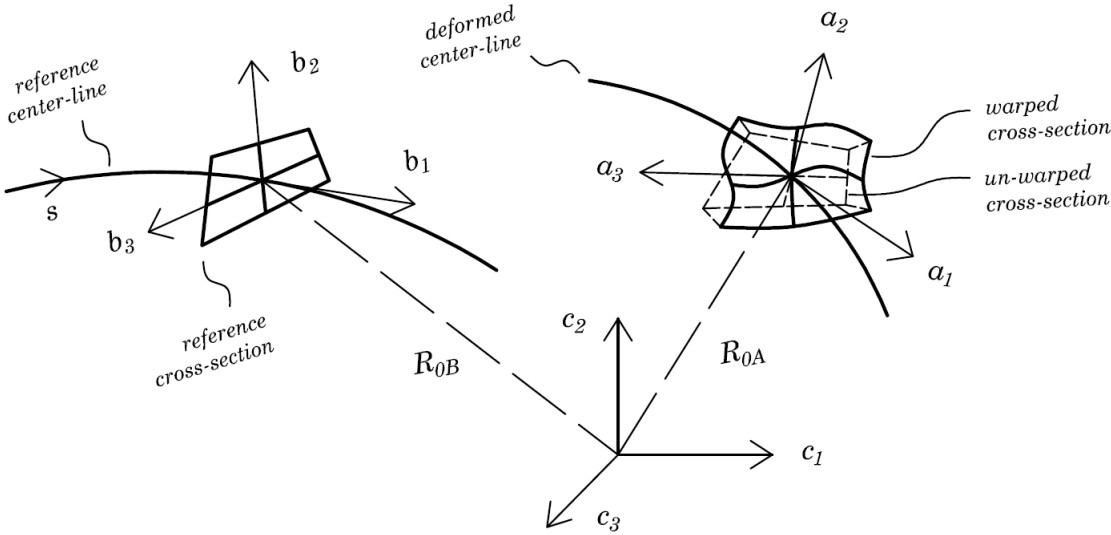

**Figure 1: Schematic of the reference and current states, centre-lines, cross-sections and local frames**

We continue by introducing the kinematical variables we use to describe the motion of the considered structure. To this aim, the orientation of the frames $a_i$ and $b_i$ relative to a fixed rectangular frame, $c_i$, are defined as follows

$$a_i = Ac_i, \quad b_i = Bc_i \tag{1}$$

where A and B are two proper orthogonal tensor fields (i.e. their determinant is 1, see, for example, Gurtin 2003). We also introduce a tensor field, T, which defines the relative orientation between the frames $a_i$ and $b_i$ as follows

$$a_i = Tb_i = AB^T b_i \tag{2}$$

We define two skew tensor fields, $K_A$ and $K_B$, and their axial vectors, $k_A$ and $k_B$, which are related to the curvatures of the centre-line of the structure in the current and reference states, as follows (see, for example, Simo 1985 and Gurtin 2003)

$$K_A = A'A^T, \quad a_i' = K_A a_i = k_A \wedge a_i$$
$$K_B = B'B^T, \quad b_i' = K_B b_i = k_B \wedge b_i \tag{3}$$

where the prime denotes derivative with respect to the arch-length, s, while the operator $\wedge$ is the usual cross-product. In a similar manner, we introduce the skew tensor field $\Omega$, and its axial vector field $\omega$, related to the variation of the vectors $a_i$ over the time, t, as follows

$$\Omega = \dot{A}A^T, \quad \dot{a}_i = \Omega a_i = \omega \wedge a_i \tag{4}$$

The dot (over the variables) denotes derivative over the time, t. At this point, it is easy to obtain the following identities

$$T'T^T = K_A - TK_B T^T, \quad \dot{T}T^T = \Omega$$
$$\phi[T'T^T] = k_A - Tk_B, \quad \phi[\dot{T}T^T] = \omega$$

(5)

where the operator $\phi[]$ provides the axial vector of the skew tensor between brackets.

The function $R_{0B}$, which maps the points of the centre-line in the reference state, does not depend on time, while $R_{0A}$ can change over the time t. Its variation is the time rate of change of the position of the points of the current centre-line

$$\dot{R}_{0A} = v_0$$

(6)

We are now in a position to introduce two important kinematic identities

$$v_0' - \omega \wedge R_{0A}' = T\dot{\gamma}$$
$$\omega' = T\dot{k}$$

(7)

where $\gamma$ and k are

$$\gamma = T^T R_{0A}' - R_{0B}'$$
$$k = T^T k_A - k_B$$

(8)

It is worth nothing that $\gamma$ and k vanish for rigid motions and are invariant under superposed rigid motion, hence, they are well-defined measures of strain for beam-like structures (see, for example, Ruta 2006 and Rubin 2000).

Now, we start modelling the motion of the points of the cross-sections. In particular, we introduce two mapping functions, $R_A$ and $R_B$, to identify the positions of the points of the 3D beam-like structure in its current and reference states. For what the reference state is concerned, we define the (reference) mapping function

$$R_B(z_i) = R_{0B}(z_1) + x_\alpha(z_i)b_\alpha(z_1)$$

(9)

where $R_{0B}$ is the position of the points of the reference centre-line relative to the frame $c_i$, $b_\alpha$ are the vectors of the reference local frame in the plane of the reference cross-section, $x_\alpha$ identify the position of the points in the reference cross-section relative to the reference centre-line, and, finally, $z_i$ are independent mathematical variables which do not depend on time. In particular, $z_1$ is equal to the arch-length s, and $z_\alpha$ belong to a bi-dimensional mathematical domain which is used to map the position of the points, $x_\alpha$, of the cross-sections. Throughout this paper, Greek indices assume values 2 and 3, Latin indices assumes values 1, 2 and 3, and repeated indices are summed over their range.

It is worth noting that $x_k$ may or may not be equal to $z_k$. The first choice leads to the most common modelling approaches available in the literature (see, for example, Simo 1985, Pai 2011, and Yu 2012). In this work we choose a set of relations between the position variables $x_k$ and the mathematical variables $z_k$ to provide a description of the shape of the considered beam-like structure, which can be curved, twisted and also tapered in its reference state. In particular, the span-wise variation of the shape of the cross-sections is analytically modelled by means of a mapping of this kind

$$x_i = \Lambda_{ij} z_j$$

(10)

where the coefficients $\Lambda_{ij}$ are functions of $z_1$. In the following we will consider the case of the curved and twisted beam-like structures with bi-tapered cross-sections, in which case the map (10) reduces to

$$x_1 = z_1, \quad x_2 = z_2 \lambda_2(z_1), \quad x_3 = z_3 \lambda_3(z_1) \tag{11}$$

where the coefficients $\lambda_\alpha$ are functions of $z_1$. It is worth noting that a suitable choice of those functions gives the possibility to reproduce interesting shapes. See, for example, Figure 2. It shows a 3D beam-like structure whose centre-line is curved, while the cross-sections are twisted and tapered from the root to tip. The reference cross-sections in Figure 2 are ellipses with different sizes and orientations, but any other reference cross-section shape can be considered, such as the aerodynamic profiles which are commonly used for wind turbine blades, steam turbines blades, and helicopter rotor blades as well (see also Griffith 2011, Bak 2013, Tanuma 2017, and Leishmain 2006 for examples of such profiles).

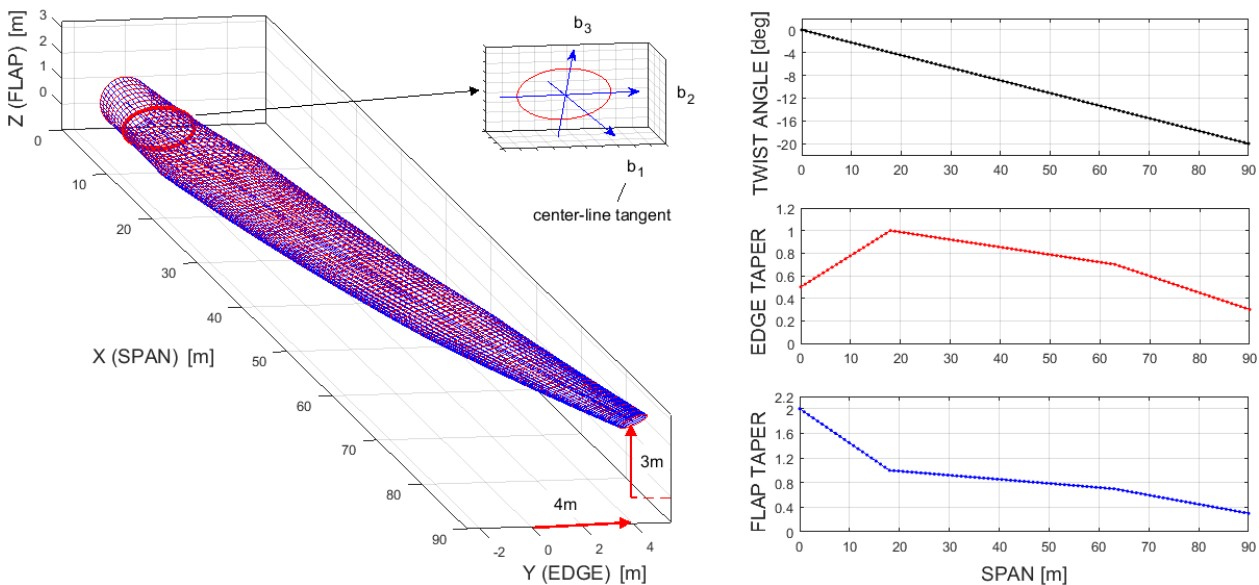

**Figure 2: Example of curved, twisted, and tapered beam-like structure and local frame (left), taper and twist functions (right)**

The position of the points in the current state are defined in a similar manner by means of the (current) mapping function

$$R_A(z_i, t) = R_{0A}(z_1, t) + x_\alpha(z_i) a_\alpha(z_1, t) + w_k(z_i, t) a_k(z_1, t) \tag{12}$$

where $R_{0A}$ is a function mapping the position of the points of the centre-line in the current state, while $w_k$ are the components of the 3D warping displacements in the local frame $a_k$. The main formal difference between the reference and current maps is due to the warping, $w$, introduced to describe the geometry of the deformed state without a-priori approximation.

By using the maps (9) and (12), we can determine the 3D tensor, $H$, expressing the gradient of the current position, $R_A$, with respect to the reference position, $R_B$, as follows (see, for example, Rubin 2000)

$$H = \frac{\partial R_A}{\partial R_B} = G_k \otimes g^k \tag{13}$$

In (13), $G_k$ and $g^k$ are covariant and controvariant base vectors, in the current and reference states, and can be calculated by using standard mathematical methods (see, for example, Rubin 2000). In this case they can be written in the form

$$
\begin{aligned}
g^1 &= g_0^{-1/2} b_1 \\
g^2 &= \Lambda_{22}^{-1}(b_2 - K_{B2\alpha}^* z_\alpha g_0^{-1/2} b_1) \\
g^3 &= \Lambda_{33}^{-1}(b_3 - K_{B3\alpha}^* z_\alpha g_0^{-1/2} b_1) \\
G_1 &= a_1 + \gamma_i a_i + K_{Ai\alpha}^* z_\alpha a_i + K_{Aij} w_j a_i + w_{i,1} a_i \\
G_2 &= \Lambda_{22} a_2 + w_{i,2} a_i \\
G_3 &= \Lambda_{33} a_3 + w_{i,3} a_i
\end{aligned}
\tag{14}
$$

where

$$
\begin{aligned}
g_0^{1/2} &= 1 + K_{B1\alpha}^* z_\alpha \\
K_{(B\,or\,A)i\alpha}^* &= \Lambda_{i\alpha}' + \Lambda_{\beta\alpha} K_{(B\,or\,A)i\beta}
\end{aligned}
\tag{15}
$$

When H is known, the 3D Green-Lagrange strain tensor, E, can be calculated (see, for example, Rubin 2000 and Gurtin 2003). Hereafter we write the tensor E in a form based on simplifying assumptions applicable to the considered beam-like structure. In particular, we introduce the characteristic dimension of the cross-sections, herein denoted as h, the longitudinal dimension of the centre-line, herein denoted as L, and we assume h to be much smaller than L. Moreover, we consider a thin structure and assume the curvatures of its reference centre-line are much smaller than 1/h (see also Rubin 2000). In addition, we assume the warping displacements, $w_k$, are small. More precisely, by introducing a non-dimensional parameter $\varepsilon$ much smaller than one, they are considered of the order of $h\varepsilon$, while the order of magnitude of their derivative with respect to $z_1$ is $\varepsilon h/L$. In general, all deformation measures, i.e. the 1D strain measures $\gamma$ and k and the components of the 3D strain tensor, E, are assumed to be small. In particular, their order of magnitude is at most $\varepsilon$. For the considered structure, in the case of small strains and small local rotations, we write the strain tensor, E, in the following form

$$
E \simeq \frac{T^T H + H^T T}{2} - I
\tag{16}
$$

Let's now calculate the components of E by using (16) and neglecting terms smaller than $\varepsilon$. Algebraic manipulations, which are based on equations (13)-(16), yields the following expressions for bi-tapered cross-sections

$$
\begin{aligned}
E_{11} &= \gamma_1 + k_2 \Lambda_{33} z_3 - k_3 \Lambda_{22} z_2 \\
E_{22} &= \Lambda_{22}^{-1} w_{2,2} \\
E_{33} &= \Lambda_{33}^{-1} w_{3,3} \\
2E_{21} &= \gamma_2 + \Lambda_{22}^{-1} w_{1,2} - k_1 \Lambda_{33} z_3 \\
2E_{31} &= \gamma_3 + \Lambda_{33}^{-1} w_{1,3} + k_1 \Lambda_{22} z_2 \\
2E_{23} &= \Lambda_{33}^{-1} w_{2,3} + \Lambda_{22}^{-1} w_{2,2}
\end{aligned}
\tag{17}
$$

In (17), $\Lambda_{22}$ and $\Lambda_{33}$ are the edge-wise and flap-wise taper coefficients (see, for example, Figure 2), while the components of the strain tensor, E, are taken with respect to the reference local frame, $b_i$, i.e.

$$E_{ij} = E \cdot b_i \otimes b_j \qquad (18)$$

where $\cdot$ is the usual scalar (or dot) product and $\otimes$ is the tensor (or dyadic) product (see, for example, Rubin 2000).

## 3.2 Stress fields and constitutive models

Given the strain tensor, E, the stress fields in the structure can be calculated when a constitutive model is chosen. Limiting our attention to elastic bodies, in a pure mechanical theory, in the case of small strain, we use the following linear relation between the second Piola-Kirchhoff stress tensor, S, and the Green-Lagrange strain tensor (see, for example, Gurtin 2003)

$$S = 2\mu E + \lambda trE \, I \qquad (19)$$

where $\mu$ and $\lambda$ are known material parameters related to the Young's modulus and Poisson's ratio. In the case of small strains and small local rotations, we also write

$$P \simeq TS, \quad C \simeq TST^T \qquad (20)$$

where P is the first Piola-Kirchhoff stress tensor and C is the Cauchy stress tensor (Gurtin 2003). It is worth noting that in the considered case the tensor field T is sufficient to determine two of the above mentioned stress tensors (e.g. P and C) when the other one (e.g. S) is known.

We are now in the position to define the cross-sectional stress resultants, namely the force F and moment M. Using the first Piola-Kirchhoff stress tensor (Gurtin 2003), in the case of small warpings, small strains and small local rotations, we write

$$F = T \int_{\Sigma} P_{i1} b_i, \quad M = T \int_{\Sigma} x_\alpha P_{i1} \, b_\alpha \wedge b_i \qquad (21)$$

where $\Sigma$ is the domain corresponding to the cross-section on which the integration is performed and

$$P_{ij} = P \cdot a_i \otimes b_j \qquad (22)$$

By combining equations (16)-(21), the force and moment stress resultants can be related to the geometrical parameters of the structure and the 1D strain measures (8). However, such relations are actually known if we know the warping fields $w_k$. An approach to obtain suitable warping fields is illustrated in section 3.4.

## 3.3 Expended power and balance equations

To complete the formulation, we conclude with considerations on the principle of expended power and the balance equations for the considered structure. For hyper-elastic bodies (Gurtin 2003), we write the principle of expended power in the form

$$\int_A p \cdot v + \int_V b \cdot v = \frac{d}{dt} \int_V \Phi \qquad (23)$$

In (23), p are surface loads per unit reference surface (A), b are body loads per unit reference volume (V), $\Phi$ is the 3D energy density function of the body, which is half the scalar product of the tensor fields S and E (i.e. $2\Phi = S \cdot E$), and, finally, v is the time rate of change of the current position of the body's points, which is given by

$$v = v_0 + \omega \wedge x_\alpha a_\alpha + \dot{w} \qquad (24)$$

where the last term in (24) is the time rate of change of the warping displacement.

For small warpings, small strains, and small local rotations, if the power expended by surface and body loads on the warping velocities is neglected, the external power, $\Pi_e$, reduces to the following form

$$\Pi_e = \Delta \left( F \cdot v_0 + M \cdot \omega \right) + \int_s F_s \cdot v_0 + M_s \cdot \omega \qquad (25)$$

where the vector field $v_0$ is the time rate of change of the position of the points of the current centre-line, the vector field $\omega$ is the time rate of change of the orientation of the vectors $a_i$, the terms $F_s$ and $M_s$ are suitable resultants of inertial actions and prescribed loads per unit length in the reference state, while the symbol $\Delta$ simply means that the function between brackets is evaluated at both the ends of the beam and the difference between those values is taken.

The cross-sectional warpings may be important in calculating the 3D energy function and cannot be neglected in the internal power, $\Pi_i$. However, the internal power may be reduced to a useful form for beam-like structures by introducing a suitable 1D strain energy function, U. For example, if U can be expressed in terms of the strain measures, $\gamma$ and k, we can write

$$\Pi_i = \frac{d}{dt} \int_s U(\gamma, k, s) = \int_s f \cdot \dot{\gamma} + m \cdot \dot{k} \qquad (26)$$

where the vector fields f and m are defined in terms of the force and moment stress resultants, F and M, as follows

$$f = T^T F, \quad m = T^T M \qquad (27)$$

By using the principle of expended power, we also obtain balance equations for the vector fields F and M in the form

$$\begin{aligned} F' + F_s &= 0 \\ M' + R'_{0A} \wedge F + M_s &= 0 \end{aligned} \qquad (28)$$

At this point, we have kinematic equations, (6)-(7), strain measures, (8) and (16), force and moment balance equations, (28), and the principle of expended power, $\Pi_e = \Pi_i$, in a suitable form for beam-like structures, (25)-(26). To complete the formulation of the model we need relations providing the 1D stress resultants in terms of the 1D strain measures. To this end, we need to know the warping fields. An approach to obtain suitable warping functions is discussed hereafter.

### 3.4 Warping displacements

In general, a 3D nonlinear elasticity problem can be formulated as a variational problem. In any case, if we try to solve the variational problem directly, the difficulties encountered in solving the elasticity problem remain. For beam-like structures whose transversal dimensions are much smaller than the longitudinal one, assumptions based on the shape of the structure and the smallness of the warping and strain fields can lead to useful simplifications. In particular, the resolution of the 3D

nonlinear elasticity problem can be reduced to the resolution of two main problems. See, for example, Berdichevsky (1981), who seems to be the first in the literature to plainly state that for elastic rods. One of those problems governs the local distortion of the cross-sections and is here referred to as the cross-sections problem. The other problem governs the global deformation of the centre-line and is here referred to as the centre-line problem. Hereafter, we consider the following variational statement to determine the warping fields which are responsible of the deformation of the cross-sections

$$\delta \int_V \Phi = 0 \tag{29}$$

In (29) the symbol $\delta$ stands for the variation operator and the density function $\Phi$ depends on the warping displacements. The warping fields satisfying (29) can be obtained by the corresponding Euler-Lagrange equations (see, for example, Courant 1953), by using numerical methods, in general, or analytical approaches providing closed-form expressions, in some particular cases. Once such a problem is solved, the components of the stress resultants (21) can be linearly related to the components of the 1D strain measures, by using equations (16)-(21). Then, if it is preferred or deemed useful, those relations can also be arranged in a standard matrix form.

Note that to determine the current state of the structure we also need the displacements of its centre-line points. They can be determined by solving the centre-line problem, which is a non-linear problem governed by the set of kinematic, constitutive and balance equations introduced in section 3 (in particular, we are referring to the constitutive model in section 3.2, which relates stress resultants and strain measures, and the balance equations for the stress resultants in section 3.3).

In the next sections we show some analytical solutions (section 4) and numerical results (section 5) that can be obtained by applying the proposed modelling approach to some reference beam-like structures.

## 4 First analytical results for bi-tapered cross-sections

In this section we consider the case of a beam-like structure with bi-tapered elliptical cross-sections. For this case we can provide analytical solutions in terms of warping fields, while for generic shapes (e.g. the aerodynamic profiles used in wind turbine blades, steam turbines blades, and helicopter rotor blades as well) the problem corresponding to (29) can be solved by using numerical methods. However, this is not surprising, since even in the classical linear theory of prismatic beams analytical solutions are available for a limited number of cases only (see, for example, Love 1944).

As discussed in section 3, we are assuming the smallness of the warpings, strains and local rotations. Moreover, hereafter we choose the current local frames to be tangent to the current centre-line and we include possible shear deformations within the warping fields. In addition, with the aim of showing a first analytical solution for bi-tapered cross-sections, in this section we neglect the effects of the initial twist of the cross-sections. Then, we write the Euler-Lagrange equations corresponding to (29), whose unknown functions are the warping fields, $w_k$. This can be done by using standard mathematical techniques (see, for example, Courant 1953). Finally, we proceed to find a solution to the resulting (partial differential equations) problem. In

particular, if we neglect the terms smaller than ε and maintain the terms related to extension, $\gamma_1$, and change of curvatures, $k_i$, the mentioned Euler-Lagrange equations are satisfied by the following warping fields

$$
\begin{aligned}
w_1 &= k_1 \frac{\rho^2 d_3^2 - d_2^2}{\rho^2 d_3^2 + d_2^2} \rho \Lambda^2 z_2 z_3 \\
w_2 &= -\nu \gamma_1 \Lambda z_2 - \nu k_2 \rho \Lambda^2 z_2 z_3 + \nu k_3 \Lambda^2 (\rho^2 z_3^2 - z_2^2)/2 \\
w_3 &= -\nu \gamma_1 \rho \Lambda z_3 + \nu k_3 \rho \Lambda^2 z_2 z_3 - \nu k_2 \Lambda^2 (\rho^2 z_3^2 - z_2^2)/2
\end{aligned}
\tag{30}
$$

where $d_2$ and $d_3$ are the major semi-axes of a reference elliptical cross-section (e.g. the one at 18m from root section in Figure 2), while $\Lambda = \Lambda_{22}$ and $\rho = \Lambda_{33}/\Lambda_{22}$ are known functions of $z_1$. Using this result, we can calculate the corresponding strain and stress fields, (16)-(20), stress resultants, (21), and strain energy function U. For example, if we consider a local frame in the reference cross-section with its origin at the cross-section's centre of mass and its axes aligned with the cross-section's principal axes of inertia (as in Figure 2), we can write the 1D strain energy function, U, in the form

$$
U = \frac{1}{2} EA \rho \Lambda^2 \gamma_1^2 + \frac{1}{2} GJ_1 \rho^2 \Lambda^4 k_1^2 + \frac{1}{2} EJ_2 \rho^3 \Lambda^4 k_2^2 + \frac{1}{2} EJ_3 \rho \Lambda^4 k_3^2
\tag{31}
$$

In (31), E is the Young modulus, G is the shear modulus, while A, $J_1$, $J_2$ and $J_3$ are the following geometrical parameters

$$
A = \pi d_2 d_3, \ \ J_1 = A d_2^2 d_3^2 / (\rho d_3^2 + \rho^{-1} d_2^2), \ \ J_2 = A d_3^2 / 4, \ \ J_3 = A d_2^2 / 4
\tag{32}
$$

An interesting result is that the initial taper appears explicitly in all the previous relations (in terms of $\rho$ and $\Lambda$). In its turn, this allows an analytical evaluation of its effects on the 3D strain fields, which can be calculated by using (17) and (30), and which are required to determine the 3D stress fields (19).

## 5 Numerical simulations

In this section we provide the results of simulations conducted by using the modelling approach discussed in section 3, which we have implemented into a numerical code in MATLAB language. Those results are also compared with the results that can be obtained by 3D-FEM simulations with the commercial software ANSYS.

In particular, we show a first set of test cases in which a beam-like structure with rectangular cross-sections undergoes large displacements, while it is fixed at one end and it is loaded at the other end by a force whose magnitude is progressively increased. In the second set of test cases we consider a more complex geometry, that is, a beam-like structure with elliptical cross-sections, which is curved, twisted and tapered in its reference configuration, while the loading condition is the same as in the first set of test cases. Finally, in the third set of test cases, we consider (four) different beam-like structures under the same loading condition. In particular, we consider a first prismatic structure with elliptical cross-sections. The second structure is a modification of the first one, on which we maintain the same cross-section at 18m from root and we add the taper according to the taper coefficients of Figure 2. Starting from this latter, we consider a third structure which includes the twist of the cross-sections, assuming the twist law of Figure 2. The fourth one is a curved, twisted and tapered structure

obtained by the third one (tapered and twisted) by adding the centre-line curvature. Then, we compare the results obtained by simulating the behaviour of these four structures to shows the effects related to their geometrical differences.

    In all the cases, the displacements of the points of the reference centre-line are calculated by solving the centre-line nonlinear problem through a numerical procedure we have implemented in MATLAB language, which is based on the kinematic, constitutive and balance equations introduced in section 3. In particular, we use the constitutive model introduced in section

3.2 to relate stress resultants and strain measures. We define the local frames orientation by using Euler angles and simulate orientation changes in terms of derivatives of those angles over the arch-length, s (see, for example, Pai 2003). We use the balance equations for the stress resultants introduced in section 3.3. Finally, we integrate (numerically) the resulting set of ordinary differential equations with respect to the arch-length, s. The results of this procedure are illustrated hereafter.

### 5.1 First set of test cases

In this set of test cases we consider a beam-like structure with rectangular cross-sections undergoing large displacements, while it is clamped at one end (i.e. the root) and it is loaded at the other end (i.e. the tip) by a force, F, whose magnitude is progressively increased (see Figure 3). The centre-line length is $d_1$=90m. The cross-section sizes are $d_2$=8m (edge-wise) and $d_3$=2m (flap-wise). The material properties are summarized by reference values of the Young's modulus, 70GPa, and Poisson's ratio, 0.25. The flap-wise tip force, F, varies from 100kN to 75000kN.

The simulations are run for different values of the tip force. The model we have implemented in MATLAB language to solve the non-linear problem provides results on the deformed configuration of the structure (e.g. Figure 3, left) within a few seconds. In all the cases, the simulation time is less than 2.4s. It is significantly less than that required by the corresponding non-linear 3D-FEM simulations carried out on the same computer, while the accuracy of the results is almost the same. A summary of the obtained results, in terms of global displacements and simulation times, is shown in Figures 3 and 4.

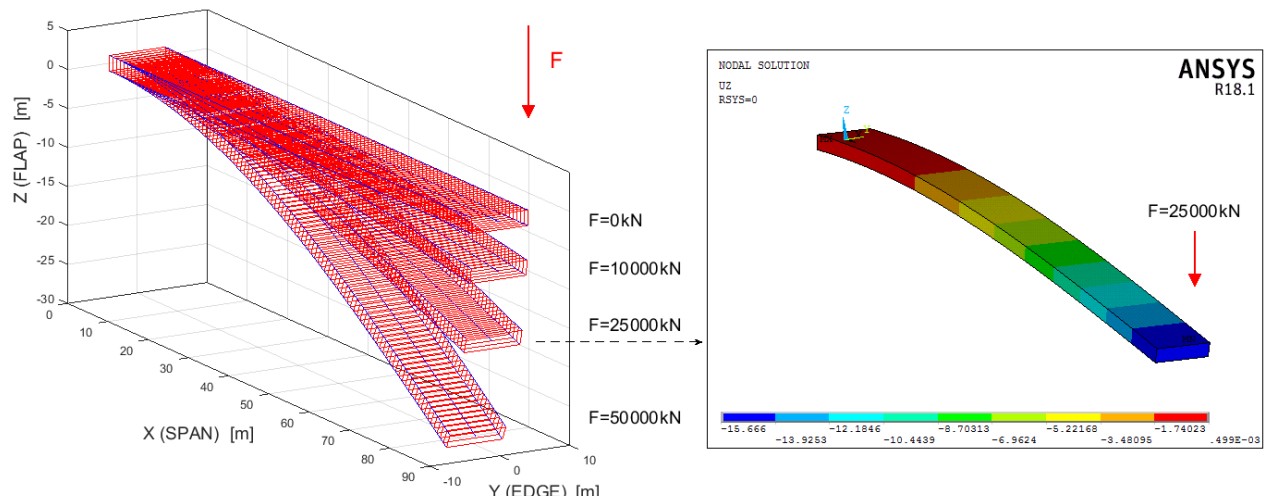

**Figure 3: Global deformation with 3D-BLM for F increasing (left) and with 3D-FEM for F=25000kN (right)**

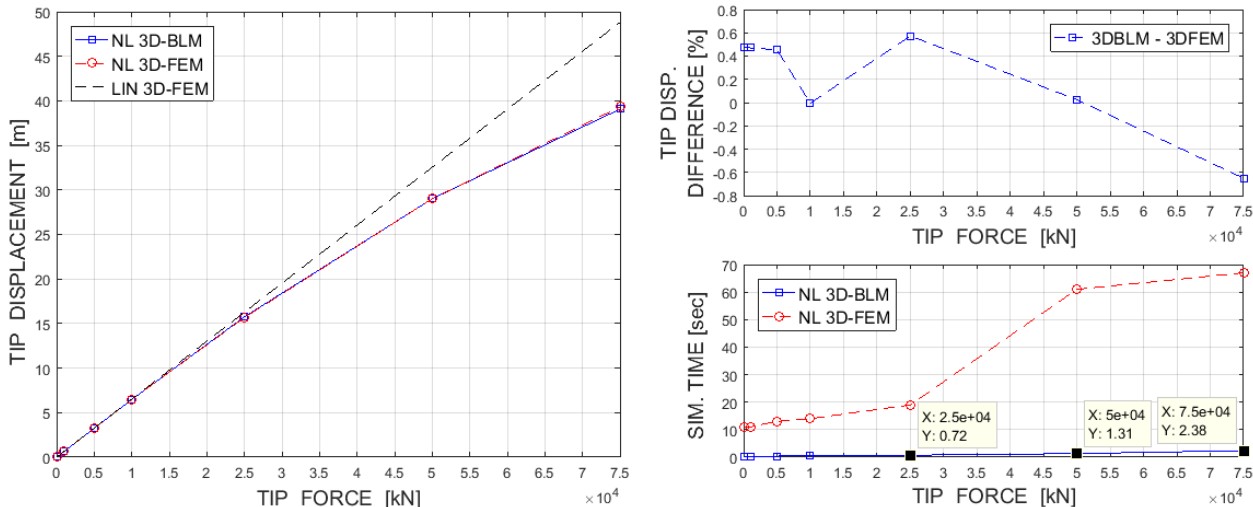

**Figure 4: Comparison of tip displacements (left), tip displacements differences and simulation times (right)**

In particular, Figure 3 (left) shows the un-deformed shape (for F=0), as well as the deformed shapes for F equal to 10000kN, 25000kN and 50000kN. Figure 3 (right) shows the 3D-FEM deformed shape for F=25000kN (the coloured legend is for the flap-wise displacements). Then, Figure 4 (left) provides a comparison between the tip displacements obtained with the linear 3D-FEM, the nonlinear 3D-FEM and our model (therein referred to as 3D-BLM). It also shows the differences (between the

non-linear 3D-FEM and the 3D-BLM) in terms of tip displacements and simulation times for the considered cases.

### 5.2 Second set of test cases

Let's now consider a more complex beam-like structure, more precisely, a 90m curved centre-line with constant curvatures, which schematizes a pre-bent and swept beam whose tip is moved 4m edgewise and 3m flap-wise, as in Figure 2. The local frames in the reference state are characterized by a pre-twist of 20deg/m. The reference cross-section at 18m from root is an

ellipse whose major semi-axes are $d_2$=2m (edge-wise) and $d_3$=0.5m (flap-wise). The sizes of the other cross-sections change according to the taper coefficients of Figure 2. For what the material properties are concerned, they are summarized by reference values of the Young's modulus, 70GPa, and Poisson's ratio, 0.25. Finally, the structure is clamped at its root and it is loaded by a flap-wise tip force, F, which varies from 100kN to 1000kN.

The simulations are run for different values of tip force, F. The model we have implemented in MATLAB language to solve

the non-linear problem provides results about the deformed configurations of the structure, such as those in Figure 5, which confirm the computational efficiency and accuracy observed in the previous section. In particular, the simulation time is significantly less than that required by corresponding nonlinear 3D-FEM simulations (see, for example, the simulation times' ratio in Figure 6, right), while the accuracy of the results is again almost the same (Figure 6).

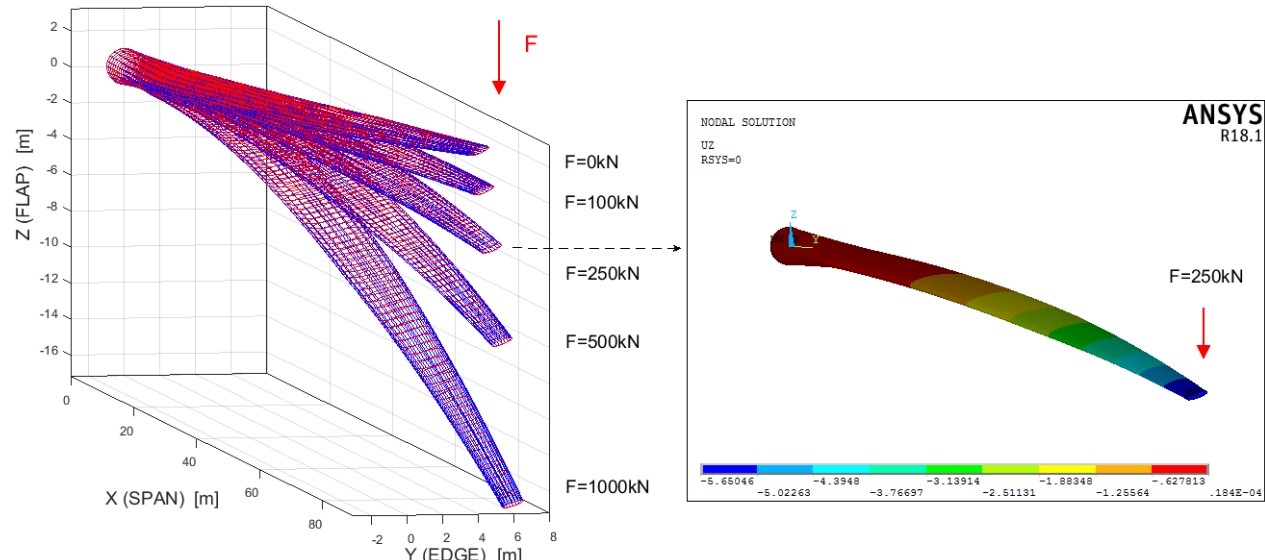

**Figure 5: Global deformation with 3D-BLM for F increasing (left) and with 3D-FEM for F=250kN (right)**

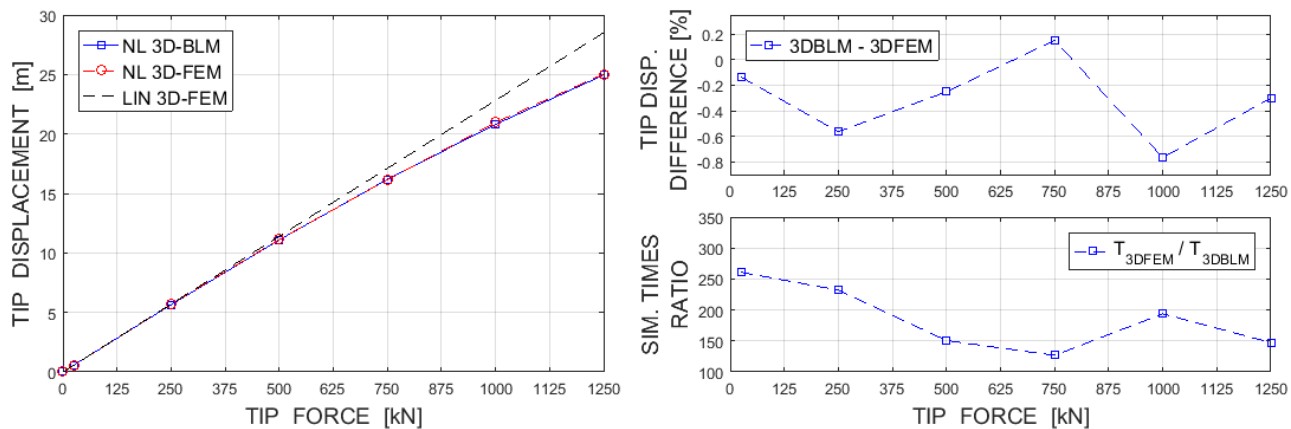

**Figure 6: Comparison of tip displacements (left), tip displacements differences and simulation times ratio (right)**

Hereafter, we continue by showing other information our model can provide. In particular, we can obtain the displacement fields of the points of the reference centre-line (Figure 7), as well as the change of curvatures of the beam-like structure (Figure 8, left) and the corresponding moment stress resultant (Figure 8, right). The moment components are in the current local frame, $a_i$, whose orientation has been determined as part of the solution of the nonlinear problem. For example, the

345 orientation of the current local frame, $a_i$, can be provided in terms of a set of Euler angles, as in Figure 9. In this case we have considered the set of Euler angles corresponding to a first rotation, $\theta$, about the initial z-axis, a second rotation, $\gamma$, about the intermediate y-axis, and a third rotation, $\psi$, about the final x-axis.

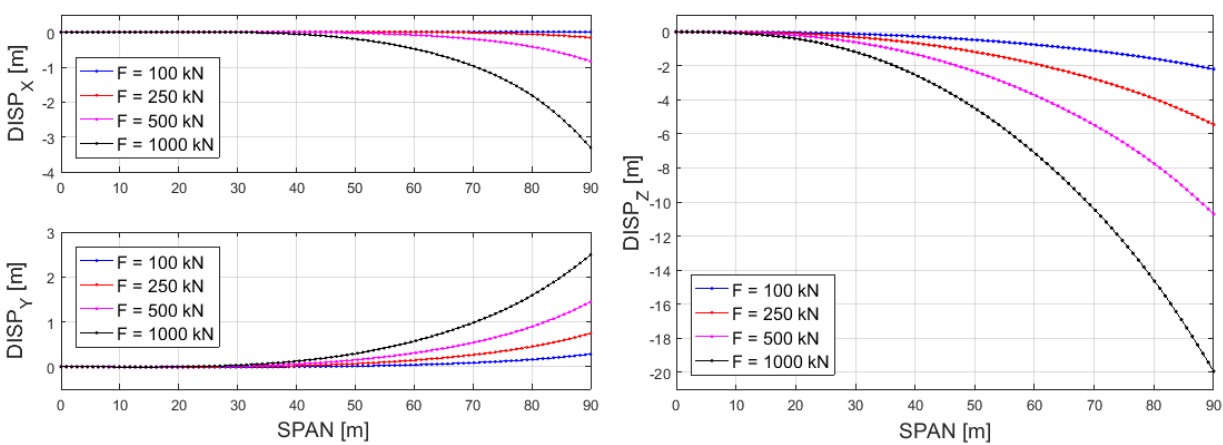

**Figure 7: Displacement of the points of the reference centre-line with 3D-BLM for F increasing**

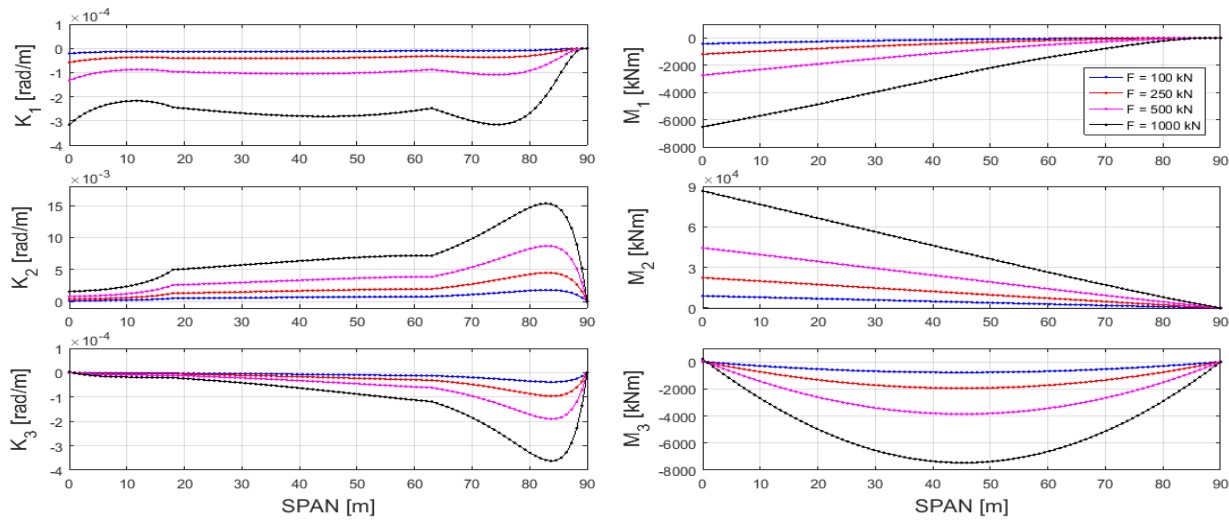

**Figure 8: Changes of curvatures (left) and moment stress resultants (right) with 3D-BLM for F increasing**

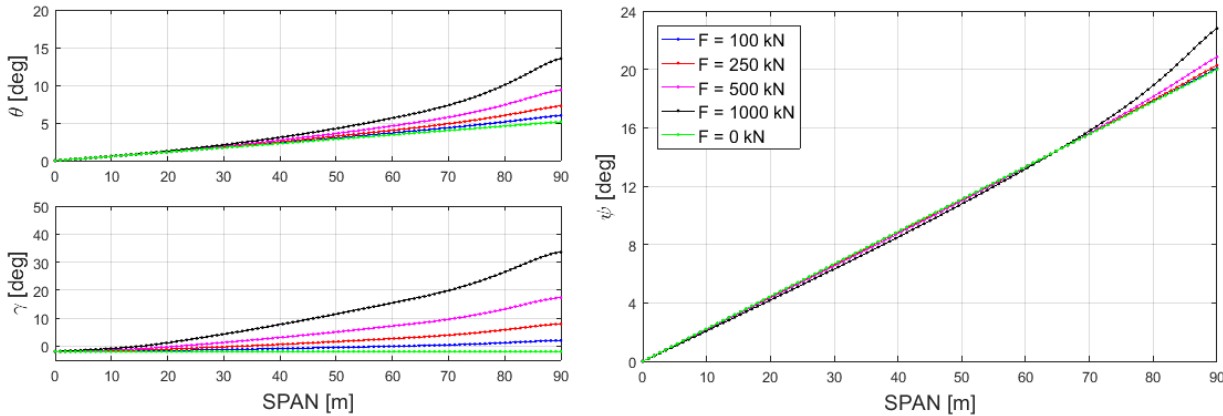

**Figure 9: Local frames orientation in terms of Euler angles before (green-lines) and after deformation**

**5.3 Third set of test cases**

Here we consider different beam-like structures, starting with a prismatic elliptical one, including step by step the taper and twist of the cross-sections and, finally, the curvature of the centre-line, as discussed in the beginning of section 5. Note that the "curved-twisted-tapered" case considered here coincides with that discussed with more details in section 5.2 (see Figures 5-9, F=250kN). We begin by simulating the behaviour of these four structures under a flap-wise tip force of 250kN. Then, we analyze the obtained results to show the effect of their geometric differences on their mechanical behaviour. A summary

of the obtained results is hereafter. In particular, Figure 10 shows the reference and deformed states of the prismatic structure (left) and the deformed states of the non-prismatic ones (right), while Figure 11 shows the displacements of their centre-lines points. The main effect of the considered tip force is a displacement in the z-direction, in all the cases, with a displacement in the y-direction that we have only for the cases "tapered-twisted" and "tapered-twisted-curved", as it is expected.

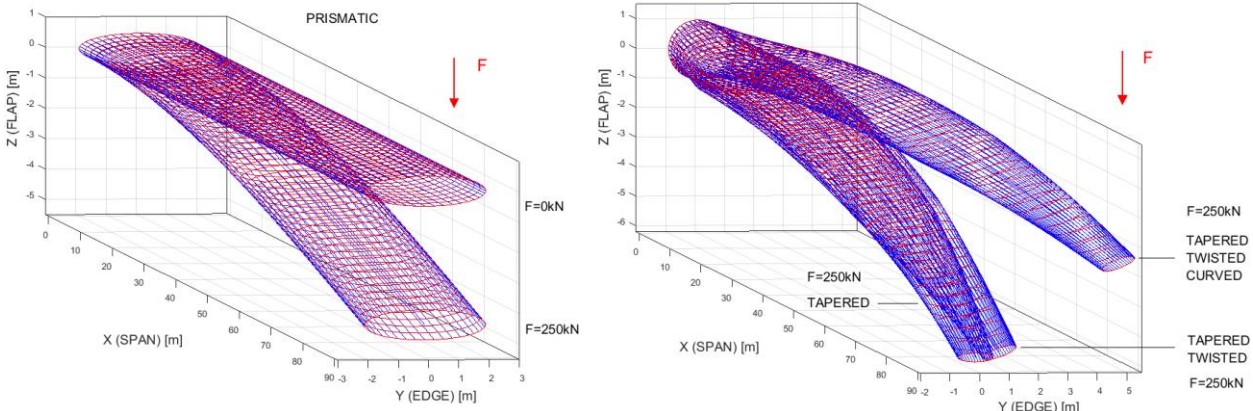

**Figure 10: Prismatic case before and after deformation (left) and non-prismatic cases after deformation (right)**

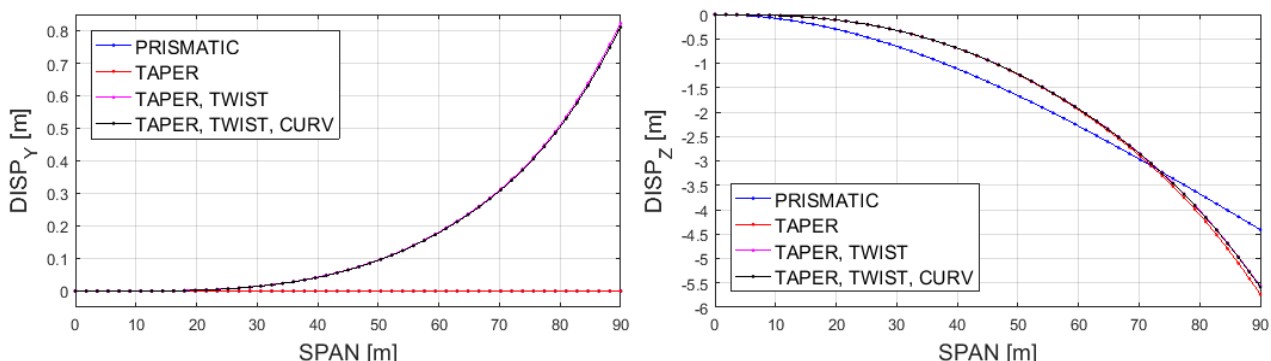

**Figure 11: Displacement of centre-line points of prismatic case (blue) and non-prismatic cases (other colours) for F=250kN**

Similar results have been obtained also for larger values of tip-force, F, which lead to larger tip-displacements. In particular, such a force, F, is varied from 250kN to 750kN. As for the previous test cases, the obtained results have been compared with

those of 3D-FEM simulations, confirming the computational efficiency and accuracy pointed out in the previous sections. A results' summary is shown in Figure 12, which provides a comparison in terms of tip-displacements, for the four geometries considered here, for F=250kN, F=500kN and F=750kN. Such loads correspond, respectively, to tip-displacements of about 6.4%, 12.5% and 18.1% of the span-wise reference length. The difference between the 3D-BLM and the non-linear 3D-FEM in terms of tip-displacements is always below 0.9% in all the considered cases (Figure 12).

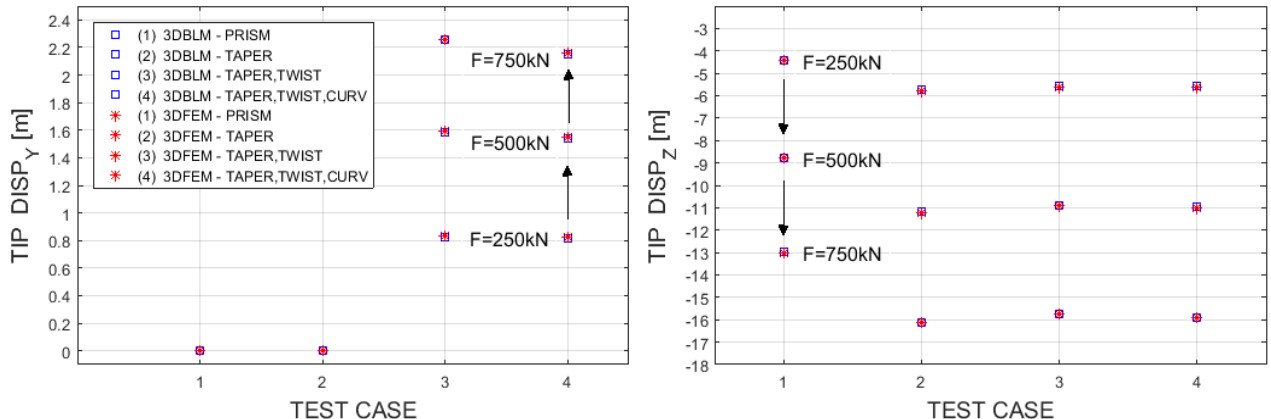

**Figure 12: Tip displacements with 3D-BLM (blue) and 3D-FEM (red) for different geometries and F increasing (see arrows)**

Hereafter, we conclude with comparison results for the 3D strain measure $E_{11}$, also referred to as longitudinal strain, which is another important parameter for the analyses and design of rotor blades (see, for example, Griffith 2011). In particular, we focus the attention on the effects of taper by considering a beam-like structure with bi-tapered cross-sections (the above "test case 2"). Then, we compare the outcomes of the 3D-BLM with those of linear and nonlinear 3D-FEM simulations. A results' summary is in Figure 13. It shows the maximum longitudinal strain at different reference cross-sections (those ones at 30%, 50%, and 70% of the span-wise reference length) and for different tip-forces (F=250kN, F=500kN and F=750N).

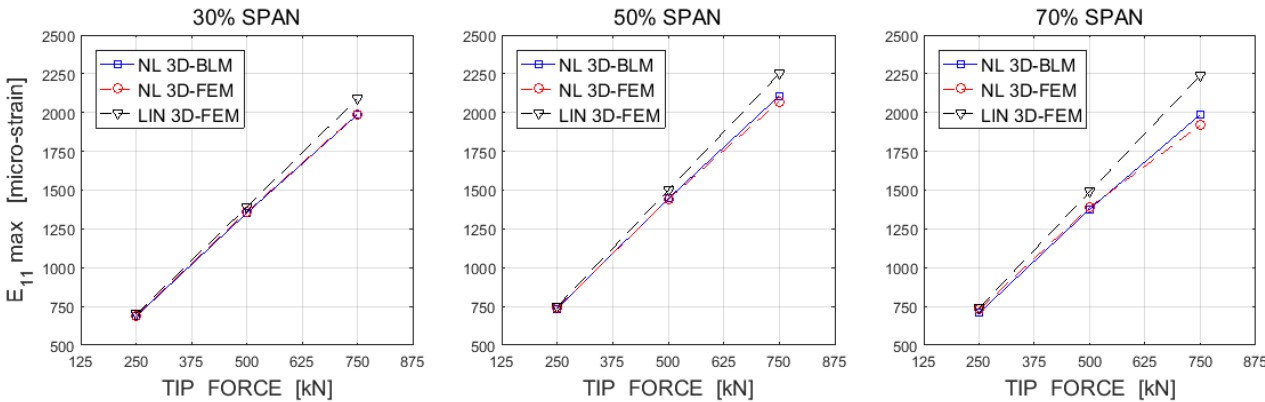

**Figure 13: Max longitudinal strain $E_{11}$ in the cross-sections at 30%, 50% and 70% span for F increasing (bi-tapered case)**

As verified by many simulations and shown in the examples, the proposed approach performs well in terms of computational efficiency and accuracy. It can be used to study the mechanical behaviour of beam-like structures, which are curved, twisted and tapered in their reference unstressed state and undergo large global displacements. It can provide information on the deformed configurations of those structures, such as their displacement fields, as well as the corresponding strain and stress measures. It is worth noting that it is suitable for beam-like structures with generic reference cross-sections shapes. However, as already pointed out, for bi-tapered elliptical cross-sections we have analytical solutions in terms of warping fields, while for generic reference cross-sections shapes the problem (29) has to be solved by using numerical methods.

## 6 Conclusions

Wind turbine blades, as well as helicopter rotor blades, steam turbine blades and many other engineering structures, can be considered complex (non-prismatic) beam-like structures, with one dimension much larger than the other two and a shape that is curved, twisted and also tapered already in the reference unstressed state. Their mechanical behaviour can be simulated through suitable 3D beam models, which are computationally efficient, accurate and explicitly consider the main geometrical design features of those structures, the large deflections of their centre-line, and the in- and out-of-plane warping of their cross-sections. In this work, curved, twisted and tapered beam-like structures have been modelled analytically. Their main geometrical characteristics have been explicitly included in the model. The warping displacement has been thought of as an additional small motion superimposed to the global motion of the local frames. The resulting model is suitable to simulate large deflections of the centre-line, large rotations of the local frames and small deformation of the cross-sections. The strain tensor has been calculated analytically in terms of the geometrical parameters of the considered structures, the 1D strain measures and the 3D warping fields. An approach based on an energy functional and a variational statement have been used to obtain suitable warping fields. The principle of expended power for curved, twisted and tapered beam-like structures has been discussed, along with the balance equations for the force and moment stress resultants. Finally, analytical results and numerical examples, which include comparison with 3D FEM simulations, have been presented to show the effectiveness of the proposed modelling approach and the information it can provide.

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
