# Peer review of "Beam-like models for the analyses of curved, twisted and tapered HAWT blades in large displacements"

_Wind Energy Science, 2019_

## Referee Comment (RC1) · Anonymous Referee #1 · 16 Sep 2019

The authors are proposing a novel beam like model specifically developed for wind turbine blade structures. The authors motivate the need for development with computational efficiency required for design optimization in conjunction with aeroelastic analysis. The model is capable of considering lengthwise geometrical variations (LGVs) such as twist, curvature and pre-bend and is suitable for large deformation analysis.

General comments: The research significance of the proposed model is high and the authors are addressing two of the renowned challenges in wind turbine blade simulations namely computational efficiency and accuracy. Regarding the latter, the implementation of LGVs into blade beam models bears indeed a considerable research

demand.

Concerning the introduction, the important contiguous contributions in the realm of this paper made by Giavotto and coworkers were not mentioned in the literature review. The model proposed in this paper is presented in a sole formal mathematical format. I am conceding the necessity of such a formal solution, albeit, the model can hardly be falsified in its current form. The authors mention that the model was indeed implemented and allude the intention to publish the procedure in a follow up paper. However, the complete absence of information concerning the implementation e.g. the pseudo code impedes reproducibility and judgement. With the information provided it is not possible to judge whether the model is a scientific breakthrough or not.

In Section 4 an analytical example is presented in which no tangible results e.g. stress/strain fields are presented that would be vital for corroboration. It would especially be pertinent (and straightforward) to compare the model predictions with analytical solutions of a tapered beam the third author published previously.

I recommend the paper for publication, provided that the solution is explicated in more detail with particular emphasis on the adopted numerical procedure. Moreover, the paper would gain credence by provision of concrete model predictions, which can be tried against analytical/other numerical solutions.

Specific comments/ questions: 1. P.2 line 40: Please define 'beam like models (BLM)' or provide a reference to its stipulation

2. P.4 line 95: Please more clearly define the meaning of 'proper orthogonal tensor fields' by preferably using a physical interpretation. The same pertains to the meaning and purpose of the skew tensor fields KA and KB. Alternatively, please provide references.

3. P.5 line 110: Please more clearly enunciate the meaning of 'well-defined measures of deformation'.
4. P.5 line 115: Please define 'proper manner'.

5. P.6 lines 150-155: The entire paragraph appears hard to follow. Can it be conflated in a more comprehensible way?

6. P.7 top: Please clearly state which higher order terms (from which order) are neglected.

7. P.7 line 170: In contrast to mathematics, I presume the majority of readers affiliated with wind energy might not be familiar with the rather specific terms stemming from differential geometry such as 'pull back' and 'push forward'. Auxiliary explanations and additional references to relevant literature would be very helpful to follow the derivation.

8. The first author of one reference is misspelled: It should rather read 'Stäblein' with umlaut.

9. P. 8 ff: Is it correct that the general beam problem is decoupled into what is stipulated as '1D' solution and into a '2D' solution? If this is indeed correctly understood, on what grounds can the decoupling be justified? What is the error estimation of such an assumption?

10. P.9 line 210: If correctly understood, the 2D solution of the warping displacements must be obtained prior to the 1D solution. Yet, in equation 28 the analytical expressions for the cross sectional properties (moments of areas) of an isotropic, prismatic ellipsoid are used. It is not abundantly clear how exactly the general 6x6 cross section stiffness matrix is obtained in case of a wind turbine rotor blade.

11. A figure showing the cross section, CSYS and cross-section forces used in section 4 would help a lot to illustrate the matter.

---

## Referee Comment (RC2) · Anonymous Referee #2 · 26 Sep 2019

The motivation of this work is highly relevant to wind energy. It is common place for beam-like models to be used, due to their balance between computational efficiency and accuracy. One limitation to these theories is the assumption of prismatic geometry. The closest example of relaxing this constraint is that of Hodges and Yu with VABS, where the beam can be curved and twisted, yet, cannot taper. Ignoring taper has some consequences for wind energy, near the root region where the loads are highest. So, the taper region can be important for structural design, while contemporary models cannot properly model these complex stresses.

Although the ambition of this work is important to wind energy, I cannot recommend

that this article is published in it's current form. A critical weakness is that the solution to the warping field is not well developed. Only a simple analytical example is given, which makes this contribution only valid for special cases. Thus, it cannot be used for wind turbine blades in general.

Currently, the state of the art are the contributions of Hodges, Yu and Giavotto. They have already developed general purpose beam models and cross section solvers. So this is the ultimate level of ambition that is needed to make a contribution to wind energy in this area. However, the key aim of this work, to incorporate taper, will be an important improvement over these earlier contributions. So I would strongly encourage the author to continue this important work.

I can recognize that getting to the level of these earlier contributions will be difficult. I think this particular manuscript can still maintain an analytical approach and be improved by expanding greatly on the example. There is still an open question on what effects a beam model with taper could capture. So, the author could demonstrate the stresses and strains that this solution gives, that are not present in a more conventional beam formulation. Furthermore, the author could also make comparisons to FEM models to highlight the effects that are not captured. This I think is possible at this level and results like this would greatly improve the manuscript. Furthermore, if you had an tapered elliptical blade, how does taper affect things like frequencies or tip deflection? Again, these results will shed light on what more we can expect from simple engineering models if this limitation was relaxed, yet although simple and analytic, it would have relevance to wind energy.

The authors did a well at explaining the motivation of their work. It could be made more widely applicable by explaining current engineering design challenges that this would help overcome. I have highlighted some points at the beginning of this review.

This is a very mathematical paper written in a concise manner, using a lot of terminology that is typically not familiar outside of the continuum-mechanics community. To

make this article accessible to wind energy readers I recommend several points where the author expand on the terminology.

The authors should further develop their techniques for solving the warping solution so it can be applied to general cross section shapes that are typically found in wind turbine blades. The authors should aim to solve the structural dynamics of real wind turbine blades. Furthermore the explanation of this work should be expanded so it is more clear.

There are several minor points that can be improved: ———————————————————————

Equation 15 with sub-equations would be more clear

A general comment as with a theoretical development, please elaborate on the assumptions taken and the limitations of this approach.

Generally speaking the wind energy community is not familiar with continuum mechanics. The author should explain verbally what all the terms mean. I personally have read about all these terms from my text books, but it would be nice if I didn't have to dust off my old texts to understand this article.

In the equations, the time rate of change is indicated by a dot. Typically this is given by a dot over the variable, however in this work it appears to be a super-script. This can be a little confusing because they use the same dot for dot products. If you use latex, \dot{x} would be the command that you would use.

The '~' operator is used in the equation. It is not clear that the '~' operator is in many of the equations. The authors should elaborate more on the formal definitions of the mathematics.

———————————————————————————

---

## Referee Comment (RC3) · Anonymous Referee #3 · 1 Oct 2019

The proposed method in the manuscript is a novel model of beam-like structures with curved, twisted and tapered geometries. Since the wind turbine blade designs are curved, twisted and tapered beam-like structures and go through large displacements in their operational life, the proposed model is highly related to the wind turbine blade analysis. Today, beam models are generally preferred in load and aeroelastic stability analysis of the turbine blades due to their accuracy and computational speed compared to the 3D finite element models. Although, curved and twisted beam models already exist in the literature (Hodges, Dewey H. Nonlinear composite beam theory), counting the taper effects are the main novelty of the study.

[Figure]

Although the motivation of the study is very interesting and notable for state of the art blade analysis, there are essential things to be done before it is published. The manuscript is written in mathematical format, however the equations are hard to follow and re-derive because authors skip intermediate steps and give no reference in the derivation of the equations. I strongly recommend to write the intermediate steps explicitly or give relevant references for these steps instead of the statements such as 'well defined measures', 'proper manner' or 'when the 2D problem is solved'. Figures depicting the cross-sectional warping effects, loads and 'suitable coordinates' (coordinate curves) would be helpful to the readers. Another substantial point is the lack of reproducible results. The analytical example results given in the manuscript can't be reproduced by the explanations given in the manuscript, hence the solution needs to be explained clearly. If the authors come up with the analytical example by themselves, they should provide more information about it. If the analytical example is taken from another study, please give reference. They should also compare the their results with a higher fidelity analysis results such as 3D finite element results to show that the taper effects are captured correctly by their formulation. The authors mention that they already implemented the method in a MatLab code. However, there is no information about the implementation of the method. Example results of authors' code and comparison of them by higher fidelity models would increase the value of the study. A wind turbine blade example would also intensify the proposed methods' relevance to wind turbine applications.

Please below suggestions:

1- Section 2 : 'BeamDyn' is very relevant to the application of the geometrically exact beam models to wind turbine analysis. Consider citing it.

2- Section 3.1 : Instead of Figure-2 with wind turbine blade, a figure with cross-section warpings and coordinate curves would be elucidating.

3- Section 3.1 : Please explain 'y' clearly (in current position vector R).

[Figure]

4- Section 3.1 : Please explain deformation gradient explicitly or give reference for it.

5- Section 3.1 : Please explain 'some higher terms' after equation 14.

6- Section 3.1 : Please write intermediate steps between equation 13 - 15.

7- Section 3.2 : A figure with cross-section forces and moments would help the readers.

8- Section 3.4 : Please elaborate the section by providing the solution of the warping fields.

9- Section 4 : Please give more information about the example and how you obtain the final results. Please, compare them with higher fidelity solution to show your model captures the taper effects correctly. Comparison can also show the results of a model which ignores the taper effects.So, reader can see the effect of taper term in final results.

10- A section which explains the numerical implementation should be added.

11- A section with results of your numerical model and higher fidelity model should be added.

---

## Author Comment (AC1) · 5 Nov 2019

| Date | 05/11/2019 |
| Our reference | WES-2019-59 |

| Contact person | Giovanni Migliaccio |
| E-mail | giovanni.migliaccio.it@gmail.com |

**Subject**   **Author's response**

University of Pisa
Department of Civil and Industrial Engineering

Address
Largo Lucio Lazzarino, 2, 56122, Pisa
Italy

Dear Reviewers,

The authors would like to express their gratitude for the constructive feedbacks which have helped us to further improve the quality of the paper. In our attempt to accounts for the comments, we have revised different parts of the paper. The objective of this document is to respond to the points raised by the Reviewers and to provide an overview of the corresponding changes that will be included in the revised paper. In the following sections, we respond to the review report provided by each Reviewer.

Your sincerely,

Giovanni Migliaccio

Sections:   Response to comments of Anonymous Referee #1
Response to comments of Anonymous Referee #2
Response to comments of Anonymous Referee #3

Note:   Author's response to each Referee's comment follows the comment itself and is in blue.

**Response to comments of Anonymous Referee #1**

The authors are proposing a novel beam like model specifically developed for wind turbine blade structures. The authors motivate the need for development with computational efficiency required for design optimization in conjunction with aeroelastic analysis. The model is capable of considering lengthwise geometrical variations (LGVs) such as twist, curvature and pre-bend and is suitable for large deformation analysis.

General comments:
The research significance of the proposed model is high and the authors are addressing two of the renowned challenges in wind turbine blade simulations namely computational efficiency and accuracy. Regarding the latter, the implementation of LGVs into blade beam models bears indeed a considerable research demand.

- Concerning the introduction, the important contiguous contributions in the realm of this paper made by Giavotto and coworkers were not mentioned in the literature review.

  Giavotto and coworkers will be cited in the literature review.

- The model proposed in this paper is presented in a sole formal mathematical format. I am conceding the necessity of such a formal solution, albeit, the model can hardly be falsified in its current form. The authors mention that the model was indeed implemented and allude the intention to publish the procedure in a follow up paper. However, the complete absence of information concerning the implementation e.g. the pseudo code impedes reproducibility and judgement. With the information provided it is not possible to judge whether the model is a scientific breakthrough or not. In Section 4 an analytical example is presented in which no tangible results e.g. stress/strain fields are presented that would be vital for corroboration. It would especially be pertinent (and straightforward) to compare the model predictions with analytical solutions of a tapered beam the third author published previously. I recommend the paper for publication, provided that the solution is explicated in more detail with particular emphasis on the adopted numerical procedure. Moreover, the paper would gain credence by provision of concrete model predictions, which can be tried against analytical/other numerical solutions.

  A section with information on the current implementation of the model in Matlab code will be provided, along with numerical results (e.g. tip deflection, strain measures, stress resultants) that can be obtained by using such a model. Comparison with corresponding results obtained with a 3D FEM commercial software will also be provided.

- Specific comments/ questions:
  1. P.2 line 40: Please define 'beam like models (BLM)' or provide a reference to its stipulation

  BLM is used as shorthand of "beam-like model". It will be better defined.

- 2. P.4 line 95: Please more clearly define the meaning of 'proper orthogonal tensor fields' by preferably using a physical interpretation. The same pertains to the meaning and purpose of the skew tensor fields KA and KB. Alternatively, please provide references.

Further details will be provided. In particular, the terms used will be better defined and more references to classical works of rational and continuum mechanics will be provided.

- 3. P.5 line 110: Please more clearly enunciate the meaning of 'well-defined measures of deformation'.

    This will be done and useful references will be provided.

- 4. P.5 line 115: Please define 'proper manner'.

    This will be done and useful references will be provided.

- 5. P.6 lines 150-155: The entire paragraph appears hard to follow. Can it be confate in a more comprehensible way?

    This paragraph will be revised and further details and references will be provided.

- 6. P.7 top: Please clearly state which higher order terms (from which order) are neglected.

    This will be better specified.

- 7. P.7 line 170: In contrast to mathematics, I presume the majority of readers affiliated with wind energy might not be familiar with the rather specific terms stemming from differential geometry such as 'pull back' and 'push forward'. Auxiliary explanations and additional references to relevant literature would be very helpful to follow the derivation.

    Auxiliary explanations and additional references to classical works of rational and continuum mechanics will be provided.

- 8. The first author of one reference is misspelled: It should rather read 'Stäblein' with umlaut.

    This will be done.

- 9. P. 8 ff: Is it correct that the general beam problem is decoupled into what is stipulated as '1D' solution and into a '2D' solution? If this is indeed correctly understood, on what grounds can the decoupling be justified? What is the error estimation of such an assumption?

    For beam-like structures with transversal dimensions much smaller than the longitudinal one, in the case discussed in section 3 (small warping, etc), the resolution of the classical three-dimensional nonlinear elasticity problem can be reduced to the resolution of two main problems. One of them governs the local warping of the cross-sections. It will be referred to as the cross-sections problem. The other one governs the global deformation of the center-line. It will be referred to as the center-line problem. In the revised version of the paper, the mathematical models to determine the deformation of cross-sections and center-line will be discussed with further details. Additional references will be provided to help understanding how those problems can be solved. A final section with

numerical simulations will be added to show the accuracy of the results obtainable with the proposed modeling approach. Comparison with corresponding results of a 3D FEM commercial software will also be provided.

- 10. P.9 line 210: If correctly understood, the 2D solution of the warping displacements must be obtained prior to the 1D solution. Yet, in equation 28 the analytical expressions for the cross sectional properties (moments of areas) of an isotropic, prismatic ellipsoid are used. It is not abundantly clear how exactly the general 6x6 cross section stiffness matrix is obtained in case of a wind turbine rotor blade.

  The analytical results proposed in section 4 are for the case of tapered (not prismatic) beam-like structures with elliptical cross-sections. For that case we can provide analytical results. For general cross-sections shapes the formulation of the problem of 'how to determine the deformation of the cross-sections' is the same (as discussed in section 3.4), but in such a case the solution has to be obtained by using a numerical method. However, this is not surprising, since even in the classical linear theory of prismatic beams the analytical solutions are available for a limited number of cases only. For what concerns the relations between the stress resultants and strain measures, they can be obtained by integration of the three-dimensional stress fields over the cross-sections of the beam-like structure. This will be better specified in the revised version of the paper.

- 11. A figure showing the cross section, CSYS and cross-section forces used in section 4 would help a lot to illustrate the matter.

  Some current figures will be modified and other figures will be added to better introduce and explain the problem. Some of them will show the cross-sections and also the local frames which are used to write the stress and strain fields in components notation.

**Response to comments of Anonymous Referee #2**

The motivation of this work is highly relevant to wind energy. It is common place for beam-like models to be used, due to their balance between computational efficiency and accuracy. One limitation to these theories is the assumption of prismatic geometry. The closest example of relaxing this constraint is that of Hodges and Yu with VABS, where the beam can be curved and twisted, yet, cannot taper. Ignoring taper has some consequences for wind energy, near the root region where the loads are highest. So, the taper region can be important for structural design, while contemporary models cannot properly model these complex stresses.

- Although the ambition of this work is important to wind energy, I cannot recommend that this article is published in it's current form. A critical weakness is that the solution to the warping field is not well developed. Only a simple analytical example is given, which makes this contribution only valid for special cases. Thus, it cannot be used for wind turbine blades in general.

  The paper addresses the modeling of the mechanical behavior of beam-like structures which are curved, twisted and tapered in their reference unstressed state, undergo large displacements, in- and out-of-plane cross-sections warping and small strain. The problem of 'how to determine the warping of the cross-sections' is formulated for generic cross-sections shapes (in section 3.4). For what concerns the resolution of such a problem, we can provide analytical results in some cases (like in section 4, for the case of tapered elliptical cross-sections) and numerical results in all the other cases. However, this is not surprising, since even in the classical linear theory of prismatic beams analytical results are available for a limited number of cases only. This will be better specified in the revised version of the paper. Moreover, an additional section with numerical results for different beam-like structures undergoing large displacements will be provided.

- Currently, the state of the art are the contributions of Hodges, Yu and Giavotto. They have already developed general purpose beam models and cross section solvers. So this is the ultimate level of ambition that is needed to make a contribution to wind energy in this area. However, the key aim of this work, to incorporate taper, will be an important improvement over these earlier contributions. So I would strongly encourage the author to continue this important work. I can recognize that getting to the level of these earlier contributions will be difficult. I think this particular manuscript can still maintain an analytical approach and be improved by expanding greatly on the example. There is still an open question on what effects a beam model with taper could capture. So, the author could demonstrate the stresses and strains that this solution gives, that are not present in a more conventional beam formulation. Furthermore, the author could also make comparisons to FEM models to highlight the effects that are not captured. This I think is possible at this level and results like this would greatly improve the manuscript. Furthermore, if you had an tapered elliptical blade, how does taper affect things like frequencies or tip deflection? Again, these results will shed light on what more we can expect from simple engineering models if this limitation was relaxed, yet although simple and analytic, it would have relevance to wind energy.

  In addition to what said in the answer to the previous comment, in the revised paper we will provide an additional section with numerical results (including tip deflection in cases

of large deflections, corresponding strain measures and stress resultants) for different reference beam-like structures. Moreover, comparison with corresponding results of a 3D FEM commercial software will also be provided.

- The authors did a well at explaining the motivation of their work. It could be made more widely applicable by explaining current engineering design challenges that this would help overcome. I have highlighted some points at the beginning of this review.
  This is a very mathematical paper written in a concise manner, using a lot of terminology that is typically not familiar outside of the continuum-mechanics community. To make this article accessible to wind energy readers I recommend several points where the author expand on the terminology.

  Auxiliary explanations as well as references to classical works of rational and continuum mechanics will be provided.

- The authors should further develop their techniques for solving the warping solution so it can be applied to general cross section shapes that are typically found in wind turbine blades. The authors should aim to solve the structural dynamics of real wind turbine blades. Furthermore the explanation of this work should be expanded so it is more clear.

  The method is already applicable to generic cross-sections shapes (see the answer to the first comment above). This will be better specified in the revised version of the paper and a section with numerical results for different test cases will be provided.

- There are several minor points that can be improved:
  Equation 15 with sub-equations would be more clear

  The corresponding paragraph will be modified and further details and references will be added.

- A general comment as with a theoretical development, please elaborate on the assumptions taken and the limitations of this approach.

  Further details about assumptions and limitations (e.g. beam-like structure, transversal dimensions much smaller than the longitudinal one, small warping, small strain, etc) will be provided and other useful references to the literature will be added.

- Generally speaking the wind energy community is not familiar with continuum mechanics. The author should explain verbally what all the terms mean. I personally have read about all these terms from my text books, but it would be nice if I didn't have to dust off my old texts to understand this article.

  Auxiliary explanations and further details will be provided. In particular, the terms used will be better defined and more references to classical works of rational and continuum mechanics will be provided to make it easier to follow the mathematical aspects of the proposed modeling approach.

- In the equations, the time rate of change is indicated by a dot. Typically this is given by a dot over the variable, however in this work it appears to be a super-script. This can be a little confusing because they use the same dot for dot products. If you use latex, ndot{x} would be the command that you would use.

This will be done, that is, the time rate of change will be indicated by a dot over the variable.

- The '^' operator is used in the equation. It is not clear that the '^' operator is in many of the equations. The authors should elaborate more on the formal definitions of the mathematics.

The operator '∧' will be better defined in the revised version of the paper.

**Response to comments of Anonymous Referee #3**

The proposed method in the manuscript is a novel model of beam-like structures with curved, twisted and tapered geometries. Since the wind turbine blade designs are curved, twisted and tapered beam-like structures and go through large displacements in their operational life, the proposed model is highly related to the wind turbine blade analysis. Today, beam models are generally preferred in load and aeroelastic stability analysis of the turbine blades due to their accuracy and computational speed compared to the 3D finite element models. Although, curved and twisted beam models already exist in the literature (Hodges, Dewey H. Nonlinear composite beam theory), counting the taper effects are the main novelty of the study.

- Although the motivation of the study is very interesting and notable for state of the art blade analysis, there are essential things to be done before it is published.
  The manuscript is written in mathematical format, however the equations are hard to follow and re-derive because authors skip intermediate steps and give no reference in the derivation of the equations. I strongly recommend to write the intermediate steps explicitly or give relevant references for these steps instead of the statements such as 'well defined measures', 'proper manner' or 'when the 2D problem is solved'. Figures depicting the cross-sectional warping effects, loads and 'suitable coordinates' (coordinate curves) would be helpful to the readers.

  Further details, explanations and references will be provided to make it easier to follow the mathematical aspects of the proposed modeling approach. Moreover, some of the current figures will be modified and other figures will be added.

- Another substantial point is the lack of reproducible results. The analytical example results given in the manuscript can't be reproduced by the explanations given in the manuscript, hence the solution needs to be explained clearly. If the authors come up with the analytical example by themselves, they should provide more information about it. If the analytical example is taken from another study, please give reference. They should also compare the their results with a higher fidelity analysis results such as 3D finite element results to show that the taper effects are captured correctly by their formulation. The authors mention that they already implemented the method in a MatLab code. However, there is no information about the implementation of the method. Example results of authors' code and comparison of them by higher fidelity models would increase the value of the study. A wind turbine blade example would also intensify the proposed methods' relevance to wind turbine applications.

  Section 4 provides analytical formulas we have obtained for beam-like structures with tapered elliptical cross-sections. This solution is also included in the code that we have implemented in Matlab. Further details on this analytical solution will be provided. Also, an additional section with numerical results (e.g. tip deflection, strain measures, stress resultants) for different beam-like structures will be added to show the effectiveness of the proposed modeling approach. Finally, comparison with corresponding results of a 3D FEM commercial software will be provided.

- Please below suggestions:
  1- Section 2 : 'BeamDyn' is very relevant to the application of the geometrically exact

beam models to wind turbine analysis. Consider citing it.

It will be cited.

- 2- Section 3.1 : Instead of Figure-2 with wind turbine blade, a figure with cross-section warpings and coordinate curves would be elucidating.

  Some figures will be modified and other figures will be added to better introduce and explain the problem.

- 3- Section 3.1 : Please explain 'γ' clearly (in current position vector R).
  4- Section 3.1 : Please explain deformation gradient explicitly or give reference for it.
  5- Section 3.1 : Please explain 'some higher terms' after equation 14.
  6- Section 3.1 : Please write intermediate steps between equation 13 - 15.

  The paragraph containing 'γ', 'deformation gradient', 'higher order terms' and 'equation 13-15' will be modified. More precisely, further details on the mathematical model will be provided, along with useful references to classical works of rational and continuum mechanics.

- 7- Section 3.2 : A figure with cross-section forces and moments would help the readers.

  Some figures will be modified and other figures will be added to better introduce and explain the problem. Some of them will show the cross-sections and also the local frames which are used to write the stress and strain fields in components notation.

- 8- Section 3.4 : Please elaborate the section by providing the solution of the warping fields.

  Section 3.4 addresses the problem of 'how to determine the warping fields', which are responsible of the cross-sections deformation. Section 4 provides analytical results for beam-like structures with tapered elliptical cross-sections. For that case we can provide analytical results. For generic cross-sections shapes the formulation of the problem is the same (as in section 3.4), but the solution has to be obtained by using numerical methods. But this is not surprising, since even in the classical linear theory of prismatic beams analytical solutions are available for a limited number of cases only. However, this will be better specified in the revised version of the paper.

- 9- Section 4 : Please give more information about the example and how you obtain the final results. Please, compare them with higher fidelity solution to show your model captures the taper effects correctly. Comparison can also show the results of a model which ignores the taper effects. So, reader can see the effect of taper term in final results.
  10- A section which explains the numerical implementation should be added.
  11- A section with results of your numerical model and higher fidelity model should be added.

  Further details on how to determine cross-sections warping and center-line deformation will be provided. An additional section will introduce the model we have implemented in Matlab and the results that it can provide. Comparison with corresponding results of a 3D FEM commercial software will also be provided.

---

## Author Response (AR1)

| Date | 30/11/2019 |
| Our reference | WES-2019-59 |

| Contact person | Giovanni Migliaccio |
| E-mail | giovanni.migliaccio.it@gmail.com |

| **Subject** | **Author's response** |

University of Pisa
Department of Civil and Industrial Engineering

Address
Largo Lucio Lazzarino, 2, 56122, Pisa
Italy

Dear Reviewers,

The authors would like to express their gratitude for the constructive feedbacks which have helped us to further improve the quality of the paper. In our attempt to accounts for the received comments, we have revised different parts of the paper. The objective of this document is to respond to the points raised by the Reviewers and to provide a detailed overview of the corresponding changes in the revised paper. In the following sections, we respond to the review report provided by each Reviewer.

Your sincerely,

Giovanni Migliaccio

Sections:   Response to comments of Anonymous Referee #1
Response to comments of Anonymous Referee #2
Response to comments of Anonymous Referee #3

Note-1:   Author's response to each Referee's comment follows the comment itself and is in blue.

Note-2:   At the end of the three sections above, a paper marked-up version is added (it provides a direct comparison between the revised paper and the initial paper).

**Response to comments of Anonymous Referee #1**

The authors are proposing a novel beam like model specifically developed for wind turbine blade structures. The authors motivate the need for development with computational efficiency required for design optimization in conjunction with aeroelastic analysis. The model is capable of considering lengthwise geometrical variations (LGVs) such as twist, curvature and pre-bend and is suitable for large deformation analysis.

General comments:
The research significance of the proposed model is high and the authors are addressing two of the renowned challenges in wind turbine blade simulations namely computational efficiency and accuracy. Regarding the latter, the implementation of LGVs into blade beam models bears indeed a considerable research demand.

- Concerning the introduction, the important contiguous contributions in the realm of this paper made by Giavotto and coworkers were not mentioned in the literature review.

  'Giavotto and coworkers' is now mentioned (see line 52 of the revised paper and the 'references').

- The model proposed in this paper is presented in a sole formal mathematical format. I am conceding the necessity of such a formal solution, albeit, the model can hardly be falsified in its current form. The authors mention that the model was indeed implemented and allude the intention to publish the procedure in a follow up paper. However, the complete absence of information concerning the implementation e.g. the pseudo code impedes reproducibility and judgement. With the information provided it is not possible to judge whether the model is a scientific breakthrough or not. In Section 4 an analytical example is presented in which no tangible results e.g. stress/strain fields are presented that would be vital for corroboration. It would especially be pertinent (and straightforward) to compare the model predictions with analytical solutions of a tapered beam the third author published previously. I recommend the paper for publication, provided that the solution is explicated in more detail with particular emphasis on the adopted numerical procedure. Moreover, the paper would gain credence by provision of concrete model predictions, which can be tried against analytical/other numerical solutions.

  An new section has been added (section 5, lines 268-367) to provide information on the current numerical implementation of the model (in Matlab), along with numerical results (e.g. tip deflections, strain measures, stress resultants) that can be obtained by using such a model (sub-sections 5.1-5.3). Comparison with corresponding results obtained with a 3D FEM commercial software are also provided.

- Specific comments/ questions:
  1. P.2 line 40: Please define 'beam like models (BLM)' or provide a reference to its stipulation

  BLM is a shorthand of "beam-like model", now it is better defined (see lines 44-46).

- 2. P.4 line 95: Please more clearly define the meaning of 'proper orthogonal tensor fields'

by preferably using a physical interpretation. The same pertains to the meaning and purpose of the skew tensor fields KA and KB. Alternatively, please provide references.

Further details have been provided (see lines 100-106) and more references to classical works of rational and continuum mechanics have been added (see line 100 for 'proper orthogonal tensor…' and line 104 for 'skew tensor…').

- 3. P.5 line 110: Please more clearly enunciate the meaning of 'well-defined measures of deformation'.
  4. P.5 line 115: Please define 'proper manner'.

  That has been done and useful references have been added (see lines 118-121).

- 5. P.6 lines 150-155: The entire paragraph appears hard to follow. Can it be confate in a more comprehensible way?

  That paragraph has been revised and more details and references have been added (see lines 157-166).

- 6. P.7 top: Please clearly state which higher order terms (from which order) are neglected.

  This has been better specified (see lines 168-169).

- 7. P.7 line 170: In contrast to mathematics, I presume the majority of readers affiliated with wind energy might not be familiar with the rather specific terms stemming from differential geometry such as 'pull back' and 'push forward'. Auxiliary explanations and additional references to relevant literature would be very helpful to follow the derivation.

  This line has been re-written and additional references to classical works of rational and continuum mechanics have been provided (see lines 181-185).

- 8. The first author of one reference is misspelled: It should rather read 'Stäblein' with umlaut.

  This has been done.

- 9. P. 8 ff: Is it correct that the general beam problem is decoupled into what is stipulated as '1D' solution and into a '2D' solution? If this is indeed correctly understood, on what grounds can the decoupling be justified? What is the error estimation of such an assumption?

  For beam-like structures with transversal dimensions much smaller than the longitudinal one, in the case discussed in section 3 (small warping, small strain, etc), the resolution of the classical 3d nonlinear elasticity problem can be reduced to the resolution of two main problems. One of them governs the local warping of the cross-sections. It is referred to as the 'cross-sections problem'. The other problem governs the global deformation of the center-line. It is referred to as the 'center-line problem'. The mathematical models to determine the deformation of cross-sections and center-line are discussed with more details in the revised paper (lines 222-243 and 283-289). Additional references have been

added to help understanding how those problems can be solved. An entire new section with numerical simulations has also been added (section 5) to show the accuracy of the results obtainable with such an approach and the information it can provide. Comparison with corresponding results of a 3D FEM commercial software have also been included.

- 10. P.9 line 210: If correctly understood, the 2D solution of the warping displacements must be obtained prior to the 1D solution. Yet, in equation 28 the analytical expressions for the cross sectional properties (moments of areas) of an isotropic, prismatic ellipsoid are used. It is not abundantly clear how exactly the general 6x6 cross section stiffness matrix is obtained in case of a wind turbine rotor blade.

  The analytical results proposed in section 4 are for the case of tapered (not prismatic) beam-like structures with elliptical cross-sections. For that case we can provide analytical results. For generic reference cross-sections shapes the formulation of the problem of 'how to determine the deformation of the cross-sections' is the same (as in section 3.4), but in such a case the solution has to be obtained by using numerical methods. However, this is not surprising, since even in the classical linear theory of prismatic beams the analytical solutions are available for a limited number of cases only (this is better specified in the revised paper, see, for example, lines 245-249). For what concerns the relations between stress resultants and strain measures, they can be obtained by integration of the 3d stress fields over the cross-sections of the beam-like structure. In the considered case they are linear relations and can be arranged in a standard matrix form (this is better specified in the revised paper, see, for example, lines 232-237).

- 11. A figure showing the cross section, CSYS and cross-section forces used in section 4 would help a lot to illustrate the matter.

  Some figures have been modified and other figures have been added to better introduce and explain the problem. In particular, see Figure 1 and Figure 2 and the corresponding lines introducing them (lines 86-95 and 142-146). They show the cross-sections and the local frames used to write the stress and strain fields, as well as the force and moments stress-resultants, in components notation. Other figures (e.g. 3, 5, 9) also help to better understand the problem and visualize the simulation results.

**Response to comments of Anonymous Referee #2**

The motivation of this work is highly relevant to wind energy. It is common place for beam-like models to be used, due to their balance between computational efficiency and accuracy. One limitation to these theories is the assumption of prismatic geometry. The closest example of relaxing this constraint is that of Hodges and Yu with VABS, where the beam can be curved and twisted, yet, cannot taper. Ignoring taper has some consequences for wind energy, near the root region where the loads are highest. So, the taper region can be important for structural design, while contemporary models cannot properly model these complex stresses.

- Although the ambition of this work is important to wind energy, I cannot recommend that this article is published in it's current form. A critical weakness is that the solution to the warping field is not well developed. Only a simple analytical example is given, which makes this contribution only valid for special cases. Thus, it cannot be used for wind turbine blades in general.

  The paper addresses the modeling of the mechanical behavior of beam-like structures which are curved, twisted and tapered in their reference unstressed state, undergo large displacements, in- and out-of-plane cross-sections warping and small strain. The problem of 'how to determine the warping of the cross-sections' is formulated for generic cross-sections shapes in section 3.4. For what concerns the resolution of the problem, we can provide analytical results in some cases (see section 4 for the case of bi-tapered elliptical cross-sections), while numerical methods are required for generic cases. But this is not surprising, since even in the classical linear theory of prismatic beams analytical results are available for a limited number of cases only. This is better specified in the revised paper, see, for example, lines 232-235 and 245-249.

- Currently, the state of the art are the contributions of Hodges, Yu and Giavotto. They have already developed general purpose beam models and cross section solvers. So this is the ultimate level of ambition that is needed to make a contribution to wind energy in this area. However, the key aim of this work, to incorporate taper, will be an important improvement over these earlier contributions. So I would strongly encourage the author to continue this important work. I can recognize that getting to the level of these earlier contributions will be difficult. I think this particular manuscript can still maintain an analytical approach and be improved by expanding greatly on the example. There is still an open question on what effects a beam model with taper could capture. So, the author could demonstrate the stresses and strains that this solution gives, that are not present in a more conventional beam formulation. Furthermore, the author could also make comparisons to FEM models to highlight the effects that are not captured. This I think is possible at this level and results like this would greatly improve the manuscript. Furthermore, if you had an tapered elliptical blade, how does taper affect things like frequencies or tip deflection? Again, these results will shed light on what more we can expect from simple engineering models if this limitation was relaxed, yet although simple and analytic, it would have relevance to wind energy.

  In addition to what said in the answer to the previous comment, in the revised paper a new section has been added, which includes numerical results (e.g. tip deflections, strain measures, stress resultants) for some reference beam-like structures undergoing large

displacements (section 5). Different test cases confirm the effectiveness of the modeling approach and illustrate the information it can provide. Comparison with corresponding results of a 3D FEM commercial software have also been included (section 5).

- The authors did a well at explaining the motivation of their work. It could be made more widely applicable by explaining current engineering design challenges that this would help overcome. I have highlighted some points at the beginning of this review.
  This is a very mathematical paper written in a concise manner, using a lot of terminology that is typically not familiar outside of the continuum-mechanics community. To make this article accessible to wind energy readers I recommend several points where the author expand on the terminology.

  Auxiliary explanations and more references to classical works of rational and continuum mechanics have been added throughout the entire paper to make it easier to follow the mathematical aspects of the proposed modeling approach.

- The authors should further develop their techniques for solving the warping solution so it can be applied to general cross section shapes that are typically found in wind turbine blades. The authors should aim to solve the structural dynamics of real wind turbine blades. Furthermore the explanation of this work should be expanded so it is more clear.

  The method is already applicable to generic cross-sections shapes (see the answer to the first comment above, page 5 of 10).

- There are several minor points that can be improved:
  Equation 15 with sub-equations would be more clear

  The corresponding lines have been revised and further details and references have been added (see lines 168-174).

- A general comment as with a theoretical development, please elaborate on the assumptions taken and the limitations of this approach.

  Further details about the assumptions (e.g. beam-like structure, transversal dimensions much smaller than longitudinal dimension, small warping, small strain) have been provided and more references to the literature have been added (see lines 158-165).

- Generally speaking the wind energy community is not familiar with continuum mechanics. The author should explain verbally what all the terms mean. I personally have read about all these terms from my text books, but it would be nice if I didn't have to dust off my old texts to understand this article.

  Auxiliary explanations, further details on the terms used, and more references to classical works of rational and continuum mechanics, have been added throughout the paper to make it easier to follow the mathematical aspects of the proposed modeling approach.

- In the equations, the time rate of change is indicated by a dot. Typically this is given by a dot over the variable, however in this work it appears to be a super-script. This can be a little confusing because they use the same dot for dot products. If you use latex, ndot{x} would be the command that you would use.

This has been done, that is, the 'time rate of change' has been indicated by a dot over the variable (see, for example, Eq. 4, line 109, as well as all the other equations in which the 'time rate of change' is used).

- The '^' operator is used in the equation. It is not clear that the '^' operator is in many of the equations. The authors should elaborate more on the formal definitions of the mathematics.

The operator '∧' has been better defined in the revised paper (see line 106).

**Response to comments of Anonymous Referee #3**

The proposed method in the manuscript is a novel model of beam-like structures with curved, twisted and tapered geometries. Since the wind turbine blade designs are curved, twisted and tapered beam-like structures and go through large displacements in their operational life, the proposed model is highly related to the wind turbine blade analysis. Today, beam models are generally preferred in load and aeroelastic stability analysis of the turbine blades due to their accuracy and computational speed compared to the 3D finite element models. Although, curved and twisted beam models already exist in the literature (Hodges, Dewey H. Nonlinear composite beam theory), counting the taper effects are the main novelty of the study.

- Although the motivation of the study is very interesting and notable for state of the art blade analysis, there are essential things to be done before it is published.
  The manuscript is written in mathematical format, however the equations are hard to follow and re-derive because authors skip intermediate steps and give no reference in the derivation of the equations. I strongly recommend to write the intermediate steps explicitly or give relevant references for these steps instead of the statements such as 'well defined measures', 'proper manner' or 'when the 2D problem is solved'. Figures depicting the cross-sectional warping effects, loads and 'suitable coordinates' (coordinate curves) would be helpful to the readers.

  Further details and references have been added throughout the paper to make it easier to follow the mathematical aspects of the proposed modeling approach. In addition, some figures have been modified to better introduce and explain the problem. See, for example, Figures 1 and 2. They show the center-lines in the reference and current states, the 'plane' cross-section in the reference state, as well as the corresponding 'warped' cross-section in the current state. Moreover, they also show the local frames which are used to write the stress and strain fields, the warping fields, as well as the force and moment stress-resultants, in components notation. Other figures also help to better understand the problem and visualize the simulation results (e.g. Figures 3, 5, and 9).

- Another substantial point is the lack of reproducible results. The analytical example results given in the manuscript can't be reproduced by the explanations given in the manuscript, hence the solution needs to be explained clearly. If the authors come up with the analytical example by themselves, they should provide more information about it. If the analytical example is taken from another study, please give reference. They should also compare the their results with a higher fidelity analysis results such as 3D finite element results to show that the taper effects are captured correctly by their formulation. The authors mention that they already implemented the method in a MatLab code. However, there is no information about the implementation of the method. Example results of authors' code and comparison of them by higher fidelity models would increase the value of the study. A wind turbine blade example would also intensify the proposed methods' relevance to wind turbine applications.

  Section 4 provides analytical formulas we have obtained for beam-like structures with bi-tapered elliptical cross-sections. More information on this analytical solution have been provided in section 3.4 and 4. Also, an entire new section with numerical results (e.g. tip deflection, strain measures, stress resultants) for different beam-like structures has been

added to show the effectiveness of the proposed approach and the information it can provide (section 5 and subsections 5.1-5.3). Comparison with corresponding results of a 3D FEM commercial software have also been included in that section.

- Please below suggestions:
  1- Section 2 : 'BeamDyn' is very relevant to the application of the geometrically exact beam models to wind turbine analysis. Consider citing it.

  It has been cited in the revised paper (see line 58 and the 'references').

- 2- Section 3.1 : Instead of Figure-2 with wind turbine blade, a figure with cross-section warpings and coordinate curves would be elucidating.

  Some figures have been modified and other figures have been added to better introduce and explain the problem. In particular, Figure 1 and Figure 2 have been modified, as well as the corresponding lines introducing them (lines 86-95 and 142-146). They show the center-lines in the reference and current states, the 'plane' cross-section in the reference state, as well as the corresponding 'warped' cross-section in the current state. Moreover, they also show the local frames which are used to write the stress and strain fields, the warping fields, and the stress-resultants, in components notation.

- 3- Section 3.1 : Please explain 'γ' clearly (in current position vector R).
  4- Section 3.1 : Please explain deformation gradient explicitly or give reference for it.
  5- Section 3.1 : Please explain 'some higher terms' after equation 14.
  6- Section 3.1 : Please write intermediate steps between equation 13 - 15.

  The paragraph containing 'γ', 'deformation gradient', 'higher order terms' and 'equation 13-15' has been modified and more explanations have been added. Further details on the mathematical model have been provided, along with more references to classical works of rational and continuum mechanics. See lines 149-174.

- 7- Section 3.2 : A figure with cross-section forces and moments would help the readers.

  Some figures have been modified and other figures have been added to better introduce and explain the problem, as mentioned above. See Figures 1-2 and the corresponding lines introducing them. They show the cross-sections and also the local frames which are used to write the stress and strain fields, the warping fields, and the force and moment stress-resultants too, in components notation.

- 8- Section 3.4 : Please elaborate the section by providing the solution of the warping fields.

  Section 3.4 addresses the problem of 'how to determine the warping fields', which are responsible of the cross-sections deformation. Section 4 provides analytical results for beam-like structures with bi-tapered elliptical cross-sections. For that case we can provide analytical results. For generic cross-sections shapes the formulation of the problem is the same as in section 3.4, but the solution has to be obtained by using numerical methods. But this is not surprising, since even in the classical linear theory of prismatic beams analytical solutions are available for a limited number of cases only. This has been better specified in the revised paper (see also lines 232-235 and 245-249).

- 9- Section 4 : Please give more information about the example and how you obtain the

final results. Please, compare them with higher fidelity solution to show your model captures the taper effects correctly. Comparison can also show the results of a model which ignores the taper effects. So, reader can see the effect of taper term in final results.

10- A section which explains the numerical implementation should be added.

11- A section with results of your numerical model and higher fidelity model should be added.

Further details on how to determine cross-sections warping and center-line deformation have been provided (see lines 222-243 and 283-289). An entire new section has been added to introduce the model we have implemented in Matlab and the results that it can provide (see section 5). Comparison with corresponding results of a 3D FEM commercial software have also been included in that section.

[revised manuscript text omitted]

---

## Author Response (AR2)

| Date | 19/02/2020 |
|---|---|
| Our reference | WES-2019-59 |
| | |
| Contact person | Giovanni Migliaccio |
| E-mail | giovanni.migliaccio.it@gmail.com |

**Subject**    **Author's response**

University of Pisa
Department of Civil and Industrial Engineering

Address
Largo Lucio Lazzarino, 2, 56122, Pisa, Italy

Dear Reviewers,

The authors would like to express their gratitude for the received comments, which have helped us to further improve the quality of the paper. In our attempt to account for them, we have revised different parts of the paper. The objective of this document is to respond to the points raised by the Reviewers and to provide a detailed overview of the corresponding changes in the revised paper. In the following sections, we respond to the review report provided by each Reviewer.

Your sincerely,

Giovanni Migliaccio

Sections:    Response to comments of Anonymous Referee #1   (round 2)
Response to comments of Anonymous Referee #3   (round 2)

Note-1:    Author's response to each Referee's comment follows the comment itself and is in blue.

Note-2:    At the end of the two sections above, a paper marked-up version is added (it provides a direct comparison between the current revised paper and its previous version).

**Response to comments of Anonymous Referee #1 (round 2)**

- The manuscript was significantly improved and the comments were addressed satisfactorily. I can recommend the manuscript for publication in Wind Energy Science.

  The authors would like to express their gratitude for the constructive feedbacks we have received since the beginning of the review process. Thanks!

**Response to comments of Anonymous Referee #3 (round 2)**

The authors propose a geometrically exact beam formulation (GEBF) which is able to the model mechanical behaviour of the structures having twist, taper and large center-line displacements. Although, GEBF is not a new method for wind turbine blade modeling, the existing models are not able to capture taper effects in strain (warping and stress) fields accurately. Wind turbine blades are long, slender, composite structures with initial twist and taper, and they go through large displacements in their operational life. Therefore, this study is highly related to wind turbine modeling and technology. Please see general comments on the new version of the manuscript.

- The authors elaborated the derivation of the equation compared to the previous version of the manuscript. They also added numerical examples including prismatic beam (5.1), blade structure (5.2) and elliptic cross-section (5.3) beam with taper, twist and curvature. They claim that the proposed method is able to capture large displacements, cross-sectional warping and small strains of curved, twisted and tapered beam-like structures. Although, existing numerical tools based on geometrically nonlinear beam theory are able to capture the curvature and twist effects accurately, the taper effects still need further research. Based on the authors' claim, the proposed method is able to capture the taper effects in strain field results. This feature is the most prominent contribution of the study. However, it is not demonstrated well in the results section. Strain results are given for analytical example only and numerical examples show only center-line displacement comparisons. Reviewer thinks, the effectiveness of the proposed model needs to be demonstrated by comparisons of strain (or warping) results with 3D FEM model, if the focus of the study is mechanical behaviour modeling of structures, which have particular geometrical characteristics such as twist, taper, in- and out-of-plane cross-section warping and large center-line displacements, as mentioned in the introduction.

  The work addresses the modeling of the mechanical behavior of non-prismatic beam-like structures, which can be curved, twisted and tapered in the reference unstressed state, undergo large displacements of the center-line's points, in- and out-of-plane warping of the cross-sections and small strain. In our attempt to account for the new comments of the reviewer, we have revised the paper accordingly. An overview of the changes in the paper, which include new comparison plots, new load cases, and comparison of strain results for a beam-like structure with bi-tapered cross-sections, is provided hereafter.

- Other points:

  1-) Authors can give comparison plots of the example in section 5.2, which shows only the beam model results in Figure 6-7-8.

  As for the test cases of section 5.1, comparison plots have now been added also for the test cases of sections 5.2. In particular, see lines 330-333 of the revised version of the paper and the current Figure 6.

- 2-) Authors use a load case with 250 kN tip load in section 5.3, and it results about 6 m tip displacement on a 90 m beam structure. This is a small displacement compared to a 90m wind turbine blade displacements occurring during its operational life. Hence, the reviewer recommends authors to use a load case which results much larger tip

displacements (around 15% - 20% of span is a good number). The difference between linear and nonlinear models are apparent after the tip displacement with 16.7% of span length as shown in figure (4).

New load cases have been added. In particular, we have added cases with larger tip-loads (more precisely, 500kN and 750kN), which correspond to larger tip-displacements, up to about 18.1% of the span-wise reference length. Comparison with the results of 3D-FEM simulations have also been added for the new load cases, as for the previous load case of 250kN. See lines 369-375 of the revised paper and the current Figure 12.

- 3-) Authors can consider adding an Appendix for long derivation or intermediate steps between equations (13-15) and warping fields of analytical example. It is not easy to go to all references (some of them are books and each one has different notations) to understand the intermediate steps. All derivation would be tracked with a consistent notation if they are given in the paper.

   The corresponding paragraphs have been revised. In particular, intermediate steps have been added between the previous equations 13-15. See lines 155-161 and 173-174 of the revised paper and equations 13-17. Further details have been added also in section 4 about the analytical example. Therein, the procedure to obtain the analytical results has been better specified. In particular, see lines 259-263, which illustrates the steps to pass from condition (29) to the solution of the corresponding mathematical problem, which is provided in equation (30). As it is stated, standard mathematical techniques have been used to write the Euler-Lagrange equations corresponding to (29). In this regard, a useful textbook is mentioned at line 261 of the revised paper (however, other mathematical textbooks may be good as well, according to what the reader prefers to read on topics such as calculus of variation and partial differential equations).

- 4-) "Rubin 1997" is cited in the manuscript but it is not in the reference list.

   This has been corrected, now it is "Rubin 2000" and it is cited in the reference list.

- 5-) Definition of "rho" is missing in equation (28) and (29).

   The definition of "rho" has been added, see line 266 of the revised paper.

- 6-) As mentioned before, the most significant feature of the proposed method is its ability to model taper effects in strain results. However, the numerical implementation of it is missing. This point should also be highlighted in all examples, showing comparisons of strain results with 3D FEM. It is not possible to say the proposed model can capture the taper effects accurately by Figure (11). The center-line displacements are so large compared to the deformations of cross-sections, so these effects cannot be seen in center line displacement results.

   Due to taper, cross-sections sizes change in span-direction, with effects on the capability of the structure to withstand applied loads. In particular, this affects the displacements of the points of the center-line of the beam-like structure and the strain fields as well. In the revised paper, we have now added also comparison of strain results with linear and nonlinear 3D-FEM simulations. See lines 380-385 and Figure 13, which specifically focus on 
[revised manuscript text omitted]

---

## Author Response (AR3)

Date     02/03/2020
Our reference     WES-2019-59

Contact person     Giovanni Migliaccio
E-mail     giovanni.migliaccio.it@gmail.com

**Subject     Author's response (iteration: minor revision)**

University of Pisa
Department of Civil and Industrial Engineering

Address
Largo Lucio Lazzarino, 2, 56122, Pisa, Italy

Dear Reviewers,

The authors would like to express their gratitude for the comments we have received during the review process, which have helped us to further improve the quality of the paper. The objective of this document is to provide an overview of the minor revisions implemented in the current version of the paper. In particular, a paper-marked up version provides a direct comparison between the current revised paper and its previous version.

Your sincerely,

Giovanni Migliaccio

[revised manuscript text omitted]

---

## Author Response (AR4)

Date    20/04/2020
Our reference    WES-2019-59

Contact person    Giovanni Migliaccio
E-mail    giovanni.migliaccio.it@gmail.com

**Subject    Author's response (minor revisions)**

University of Pisa
Department of Civil and Industrial Engineering

Address
Largo Lucio Lazzarino, 2, 56122, Pisa, Italy

Dear Editor,

The authors would like to express their gratitude for the comments we have received, which have helped us to further improve the quality of the paper. The objective of this document is to provide an overview of the minor revisions implemented in the current version of the paper. A paper-marked up version provides a direct comparison between the current version of the paper and its previous version.

Your sincerely,

Giovanni Migliaccio

Section(s):    Response to Editor's comments (minor revisions)

Note-1:    Author's response to Editor's comment follows the comment itself and is in blue.

Note-2:    At the end of the section above, a paper marked-up version is added (it provides a direct comparison between the current revised paper and its previous version).

**Response to Editor's comments (minor revisions)**

Comments to the Author:

Nice job addressing the substantive comments from the reviewers. See detailed notes for additional considerations for improvement prior to publication.

- Some minor comments for final clean-up before publication:
  Line 31 – bridge components
  Line 32 – research
  Line 33 – need comma after paper

  Apart for some minor changes (e.g. for clean-up as suggested above), the present version of the paper includes modifications based on the comments received on "Section 2" and "Section 6", as detailed hereafter.

- Section 2

  The main paragraphs of section 2 have been modified. In particular, its first paragraph in the revised paper (lines 39-54) now includes further information and citations to reviews on blades aero-elastic modeling, along with references to fast aerodynamic approaches (such as BEM) and multi-objective optimization tasks (e.g. the integrated optimization of aerodynamics, structural performance and control system behavior). Then, the attention is dedicated to structural approaches for complex beam-like structures (to which modern blades belong), which is the focus of this paper (lines 55-73). More precisely, some of the main modeling approaches and investigators have been mentioned, along with literature reviews which summarize available theories and complicating effects (e.g. non-uniform cross-sections, initial twist, taper, swept), pointing out the need for further investigation on models for complex non-prismatic cases. Finally, guidelines considered in this work to develop a mathematical model for complex non-prismatic structures (undergoing large deflections, 3d cross-sectional warprings, etc) have been introduced (lines 74-80).

- Section 6

  Overall conclusion has been broadened and reorganized. In particular, after recalling the objective of the present work (lines 411-416), the advantages of the proposed modeling approach (with respect to 3D-FEM) for structural modeling of non-prismatic beam-like structures have been highlighted (e.g. lines 420-425), as well as the assumption of small warpings and strains considered in this paper and the limitation of the analyses to the terms of order $\varepsilon$ (lines 426-428). Finally, the next steps of the present research have been anticipated, e.g. the extension of the theoretical model by including higher order terms (lines 429-431), along with suggestions for other future activities of practical interest (see also lines 431-434), e.g. comparison analyses with other structural models not based on 3D-FEM (with the aim of assessing the performance of different models for non-prismatic beam-like structures in terms of information each approach can provide, computational efficiency and results accuracy).

**Beam-like models for the analyses of curved, twisted and tapered HAWT blades in large displacements**

Giovanni Migliaccio[1], Giuseppe Ruta[2], Stefano Bennati[1], Riccardo Barsotti[1]

[1]Civil and Industrial Engineering, University of Pisa, Pisa, 56122, Italy
[2]Structural and Geotechnical Engineering, University "La Sapienza" of Roma, Roma, I-00184, Italy

*Correspondence to*: Giovanni Migliaccio (giovanni.migliaccio.it@gmail.com)

**Abstract.** Continuous ongoing efforts to better predict the mechanical behaviour of complex beam-like structures , such as wind turbine blades, are motivated by the need to improve their performance  and reduce the costs. However, new design approaches and the increasing flexibility of such structures make their aero-elastic modelling ever more challenging. For the structural part of this modelling, the best compromise between computational efficiency and accuracy can be obtained via schematizations based on suitable beam-like elements. This paper addresses the modelling of the mechanical behaviour of beam-like structures which are curved, twisted and tapered in their  unstressed state and undergo large displacements, in- and out-of-plane cross-sectional warping and small strains. A suitable model for the problem at hand is proposed. Analytical and numerical results obtained by its application are presented and compared with results from 3D-FEM analyses.

**1 Introduction**

New methods are continuously being sought to improve the performance and efficiency of  horizontal axis wind turbines (HAWT). Specifically, such improvements aim to increase their energy capture capacity, develop more reliable structures, and lower overall energy costs (Wiser 2016). Such goals can be achieved through the use of advanced materials, the optimization of the aerodynamic and structural behaviour of the blades, and the exploitation of load control techniques (see, for example, Ashwill 2010, Bottasso 2012, Stäblein 2017). However, new design strategies and the increasing flexibility of those structures make  modelling  their aero-elastic behaviour ever more challenging. For the structural part of this modelling, schematizing the blades through suitable beam-like elements may represent the best compromise between computational efficiency and accuracy. Modern blades are however very complex beam-like structures. They may be curved, twisted and also tapered in their unstressed state. Even ignoring the complexities related to the materials and loading conditions, their shape alone is sufficient to make  mathematical description of their mechanical behaviour a very challenging task. This work addresses the modelling of the mechanical behaviour of

30 structures of this kind, with a particular focus on their main geometrical characteristics, such as the twist and taper of the transversal cross-sections, as well as the in- and out-of-plane cross-sectional warping  and the large deflections of their reference centre-line.

Over the years several theories have been developed for beam-like structures (see, for example, Love 1944, Antmann 1966,  Rubin 1997), for applications in different fields, from

35 helicopter rotor blades in aerospace engineering to bridge components in civil engineering and surgical tools in medicine. Nevertheless, due to the continuous need for ever more rigorous and application-oriented models, interest in advanced theories for complex beam-like structures has led to further research even in recent years. The focus of this paper is on the effects of important geometrical characteristics of those structures, such as the curvature of their centre-line, as well as

40 the twist and the taper of their cross-sections. After an introduction to modelling approaches for structures of this kind (section 2), a suitable model is proposed for the problem at hand  (section 3). Finally, analytical results and numerical examples obtained by applying the proposed modelling approach to reference beam-like structures are presented and compared with results from 3D-FEM analyses (sections 4 and 5).

**2 Overview of modelling approaches**

45 Modelling the mechanical behaviour of modern blades can be performed via different approaches . See, for example, the reviews on aero-elastic modelling approaches for wind turbine blades of Hansen 2006 and Wang 2016a, which discuss and compare aerodynamic and structural models used in research and industrial applications. For the structural modelling, two main choices are based on 3D FEM and beam models . In general, 3D FEM approaches can be very

50 accurate and flexible, but they can be computationally demanding for analyses of complex systems, especially if they are coupled with CFD methods for aerodynamic analyses . The overall computational cost can be reduced using faster aerodynamic models , such as those based on the blade element momentum theory (see, for example, Hansen 2006). However, this may not yet be sufficient in the case of multi-objective optimization tasks, in which the

55 optimization of several aspects (e.g. aerodynamic performance, structural characteristics and control systems) have to be addressed at the same time (see also Bottasso 2012). Therefore, faster structural models may be needed as well, such as suitable beam models , which may provide accurate information on the deflection of the structure's centre-line, as well as the strain and stress fields in the three-dimensional solid. The use of fast aerodynamic models along with suitable beam models may then represent the best compromise between computational efficiency and accuracy. In this work , the

60 focus is only on the structural modelling. In particular, a mathematical model is proposed to simulate the behaviour of non-prismatic beam-like structures

BLM), which may be curved, twisted and tapered in their unstressed reference state, undergo large deflections, in- and out-of-plane cross-sectional warping and small strain (such a model is referred to here as beam-like model or BLM).

Over the years many approach have been developed for beam-like structures, from classical beam models (Love 1944), for extension, twisting and bending, to Reissner's formulation (1981), which also accounts for transverse shear deformation, to geometrically exact and asymptotic approaches, involving the research efforts of many investigators (such as Antman 1966, Giavotto 1983, Simo 1985, Ibrahimbegovic 1995, Ruta 2006, Pai 2011, Yu 2012, Hodges 2018). The available theories may be broadly grouped into engineering theories and mathematical ones. The former are usually based on ad-hoc corrections to simpler theories (e.g. Rosen 1978) or exploit geometrically exact approaches (such as Wang 2016b), the latter are generally based on the directed continuum (see, for example, Rubin 2000) or exploit asymptotic methods (e.g. Yu 2012). Reviews on beam theories are also available in the literature, which summarize modelling approaches and complicating effects. For example, many theories have been developed for helicopter rotor blades with an initial twist (Hodges 1990). In this regards, a wide-ranging review on pre-twisted rods is due to Rosen (1991), which covers several aspects of the problem, from the response to static loads, to dynamics and stability issues. Kunz (1994) also provided an overview on modelling methods for rotating beams, discussing how engineering theories for rotor blades evolved over the years, from the recognition of the importance of bending flexibility, to the development of linear equations for bending and torsion, to the introduction of nonlinear terms to such equations. More recently, Rafiee (2017) reviewed the vibrations control issues in rotating beams, summarizing beam theories and complicating effects, such as non-uniform cross-sections, initial curvature, twist and sweep. In general, it seems that, unlike the case of pre-twisted rods, the results published for curved rotating beams with initial taper and sweep are quite scarce, although all these geometrical characteristics may play an important role. This is particularly true for modern wind

95  turbine blades, which are ever more flexible and longer than the past, pre-bent and swept and, in addition, are characterized by significant chord and twist variations.

 To date many research efforts have been devoted to developing powerful theories for beam-like

100  structures. However, complex non-prismatic cases still require further investigation. In general, the geometry of the reference and current states of the structure must be appropriately described, as the curvature, twist and taper are important geometrical design features and should be explicitly included in the model. Moreover, the analysis should not be restricted to small displacements. The model should provide the stress and strain  fields in the three-dimensional solid, be rigorous and application-oriented, and provide classical results when applied to prismatic cases. Following these guidelines, a mathematical model to simulate the

105  mechanical behaviour of the mentioned non-prismatic beam-like structures is proposed hereafter.

**3 Mechanical model for complex beam-like structures**

Here we are concerned with developing a mathematical model to describe the mechanical behaviour of beam-like structures which are curved, twisted and tapered in their reference state and undergo large displacements. One of the main issues with such a task is how to describe the motion of the structure . See, among others, the works of Simo 1985,

110  Ruta 2006 and Pai 2014 for some examples of different approaches. Here, we consider a non-prismatic beam-like structure as a collection of deformable plane figures (i.e. the reference cross-sections) along a suitable three-dimensional curve (i.e. the reference centre-line). We assume that each point of each cross-section in the reference state moves to its position in the current state through a global rigid motion on which a local general (warping) motion is superimposed. In this manner, the cross-sectional deformation can be examined

115  independently of the global motion of the centre-line. It is thus possible to consider the global motion to be large, while the local motion and the strain may be small. An analytical description of how the motion of the considered structure is modelled in this work is presented and further discussed in the following section.

[revised manuscript text omitted]
 steam turbines blades, as well as helicopter rotor blades) the problem corresponding to (29) can be solved using numerical methods. However, this is not surprising, since analytical solutions are available only for a limited number of cases even in the classical linear theory of prismatic beams (see, for example, Love 1944).

As discussed in section 3, we are assuming that the warpings, strains and local rotations are small. Moreover, hereafter we choose that the current local frames be tangent to the current centre-line and include possible shear deformations within the warping fields. In addition, with the aim of showing a first analytical solution for bi-tapered cross-sections, here we neglect the effects of the initial cross-sectional twist. Then, we write the Euler-Lagrange equations corresponding to (29), whose unknown functions are the warping fields, $w_k$ (this can be done using standard mathematical techniques, see, for example, Courant 1953). Finally, we proceed to find a solution to the resulting (partial differential equations) problem. In particular, if we neglect the terms smaller than $\varepsilon$ and maintain those related to extension, $\gamma_1$, and changes in curvature, $k_i$, the aforementioned Euler-Lagrange equations are satisfied by the following warping fields

$$
\begin{aligned}
w_1 &= k_1 \frac{\rho^2 d_3^2 - d_2^2}{\rho^2 d_3^2 + d_2^2} \rho \Lambda^2 z_2 z_3 \\
w_2 &= -\nu \gamma_1 \Lambda z_2 - \nu k_2 \rho \Lambda^2 z_2 z_3 + \nu k_3 \Lambda^2 (\rho^2 z_3^2 - z_2^2)/2 \\
w_3 &= -\nu \gamma_1 \rho \Lambda z_3 + \nu k_3 \rho \Lambda^2 z_2 z_3 - \nu k_2 \Lambda^2 (\rho^2 z_3^2 - z_2^2)/2
\end{aligned}
\tag{30}
$$

where $d_2$ and $d_3$ are the major semi-axes of a reference elliptical cross-section (e.g. the one at 18m from the root section in Figure 2), while $\Lambda = \Lambda_{22}$ and $\rho = \Lambda_{33}/\Lambda_{22}$ are known functions of $z_1$. Using this result, we can calculate the corresponding strain and stress fields, (16)-(20), stress resultants, (21), and strain energy function U. For example, if we consider a local frame in the reference cross-section with its origin at the cross-section's centre of mass and its axes aligned with the cross-section's principal axes of inertia (as in Figure 2), we can write the 1D strain energy function, U, in the form

$$
U = \frac{1}{2} EA\rho \Lambda^2 \gamma_1^2 + \frac{1}{2} GJ_1 \rho^2 \Lambda^4 k_1^2 + \frac{1}{2} EJ_2 \rho^3 \Lambda^4 k_2^2 + \frac{1}{2} EJ_3 \rho \Lambda^4 k_3^2
\tag{31}
$$

In (31), E is the Young modulus, G is the shear modulus, while A, $J_1$, $J_2$ and $J_3$ are the following geometrical parameters

$$
A = \pi d_2 d_3, \quad J_1 = A d_2^2 d_3^2 / (\rho d_3^2 + \rho^{-1} d_2^2), \quad J_2 = A d_3^2 / 4, \quad J_3 = A d_2^2 / 4
\tag{32}
$$

An interesting result is that the initial taper appears explicitly in all the previous relations (in terms of $\rho$ and $\Lambda$). This, in turn, allows analytical evaluation of its effects on the 3D strain fields, which can be calculated by using (17) and (30) and are, in any case, required to determine the 3D stress fields (19).

**5 Numerical simulations**

In this section we present the results of simulations conducted  using the modelling approach discussed in section 3, which we have implemented into a numerical code in MATLAB language. The results are also compared with those obtainable via 3D-FEM simulations with the commercial software ANSYS.

In particular, we show a first set of test cases in which a beam-like structure with rectangular cross-sections undergoes large displacements, while fixed at one end and loaded at the other by a force  of progressively increasing magnitude. The second set of test cases addresses a more complex geometry, that is, a beam-like structure with elliptical cross-sections, which is curved, twisted and tapered in its reference configuration, under the same loading condition  as in the first set of test cases. Finally, the third set of test cases regards four different beam-like structures under the same loading conditions. In particular, we consider a first prismatic structure with elliptical cross-sections. The second structure is a modification of the first , on which  the same cross-section is maintained at 18m from  root , while taper is added according to the taper coefficients in Figure 2. Starting with this latter, we then consider a third structure which includes twisting of the cross-sections, assuming the twist law in Figure 2. The fourth and final case is a curved, twisted and tapered structure obtained from the third  (tapered and twisted) by adding a centre-line curvature. Once the simulations have been completed, we compare the results obtained  to highlight the effects  of their different geometries on their mechanical behaviour.

In all cases, the displacements of the reference centre-line points are calculated by solving the centre-line nonlinear problem through the previously mentioned numerical procedure we have implemented in MATLAB , which is based on the kinematic, constitutive and balance equations introduced in section 3. In particular, the constitutive model introduced in section 3.2 is used to relate stress resultants and strain measures. We define the local frames orientation  using Euler angles and simulate orientation changes in terms of the derivatives of those angles over the arch-length, s (see, for example, Pai 2003). We use the balance equations for the stress resultants introduced in section 3.3. Finally,  the resulting set of ordinary differential equations is (numerically) integrated with respect to the reference arch-length, s. The results of this procedure are illustrated in the following sections.

**5.1 First set of test cases**

In this set of test cases we consider a rectangular cross-sectioned beam-like structure which undergoes large displacements, while clamped at one end (i.e. the root) and loaded at the other  (i.e. the tip) by a force, F, whose magnitude is progressively increased (see Figure 3). The centre-line length is $d_1=90$m , while the cross-section dimensions are $d_2=8$m (edge-wise) and $d_3=2$m (flap-wise). The material properties are

summarized by reference values of  Young's modulus, 70GPa, and Poisson's ratio, 0.25. Finally, the flap-wise tip force, F, varies from 100kN to 75000kN.

The simulations are run for different values of the tip force. The model we have implemented in MATLAB for solving the non-linear problem renders results on the structure's deformed configuration  (e.g. Figure 3, left) within a few seconds. In all  cases, the simulation time is less than 2.4s, which is significantly less than that required for the corresponding non-linear 3D-FEM simulations carried out on the same computer, while the accuracy of the results is almost the same. A summary of the results obtained , in terms of global displacements and simulation times, is shown in Figures 3 and 4.

[Figure]

**Figure 3: Global deformation with 3D-BLM for  increasing F (left) and with 3D-FEM for F=25000kN (right)**

[Figure]

**Figure 4: Comparison of tip displacements (left), tip displacement differences and simulation times (right)**

365

In particular, Figure 3 (left) shows the un-deformed shape (for F=0), as well as the deformed shapes for F equal to 10000kN, 25000kN and 50000kN. Figure 3 (right) shows the 3D-FEM deformed shape for F=25000kN (the coloured scale is for the flap-wise displacements).  Figure 4 (left) provides a comparison between the tip displacements obtained with the linear 3D-FEM, the nonlinear 3D-FEM and our model (indicated as 3D-BLM). It also shows the

370 differences (between the non-linear 3D-FEM and the 3D-BLM) in terms of tip displacements and simulation times for the considered cases.

**5.2 Second set of test cases**

Let's now consider a more complex beam-like structure, specifically, one with a 90m curved centre-line with constant curvatures, which schematizes a pre-bent and swept beam whose tip is moved 4m edgewise and 3m flap-wise, as in

375 Figure 2. The local frames in the reference state are characterized by a pre-twist of 20deg/m. The reference cross-section at 18m from the root is an ellipse whose major semi-axes are $d_2$=2m (edge-wise) and $d_3$=0.5m (flap-wise). The sizes of the other cross-sections change according to the taper coefficients in Figure 2. The material properties are represented by reference values of  Young's modulus, 70GPa, and Poisson's ratio, 0.25. Finally, the structure is clamped at its root and loaded by a flap-wise tip force, F, which varies from 100kN to 1000kN.

380 The simulations are run for different values of tip force, F. The model we have implemented in MATLAB for solving the non-linear problem yields results regarding the structure's deformed configurations , such as those in Figure 5, which confirm the computational efficiency and accuracy observed in the previous tests. In particular,  simulation times are significantly shorter than those required by corresponding

nonlinear 3D-FEM simulations (see, for example, the simulation times' ratio in Figure 6, right), while the accuracy of the results is again nearly the same (Figure 6).

[Figure]

**Figure 5: Global deformation with 3D-BLM for increasing F (left) and with 3D-FEM for F=250kN (right)**

[Figure]

**Figure 6: Comparison of tip displacements (left), tip  displacement differences and simulation times ratio (right)**

Apart from the foregoing results, the model is also able to provide other meaningful information . In particular, we can obtain the displacement fields of the  reference centre-line points (Figure 7), as well as the change in curvature of the beam-like structure (Figure 8, left) and the corresponding moment stress resultant (Figure 8, right). The moment components are in the current local frame, $a_i$, whose

orientation has been determined as part of the solution to the nonlinear problem. For example, the orientation of the current local frame, $a_i$, can be expressed in terms of a set of Euler angles, as in Figure 9. In this case we have considered the set of Euler angles corresponding to a first rotation, $\theta$, about the initial z-axis, a second rotation, $\gamma$, about the intermediate y-axis, and a third rotation, $\psi$, about the final x-axis.

[Figure]

**Figure 7: Displacement of the  reference centre-line points with 3D-BLM for  increasing F**

[Figure]

**Figure 8: Changes in curvature (left) and moment stress resultants (right) with 3D-BLM for  increasing F**

[Figure]

**Figure 9: Local frames orientation in terms of Euler angles before (green-lines) and after deformation**

410 **5.3 Third set of test cases**

The last test cases regard four different beam-like structures, starting with a prismatic elliptical one,  to which there is the stepbystep addition of the taper , the twist of the cross-sections and, finally, the curvature of the centre-line, as discussed in  section 5. Note that the "curved-twisted-tapered" case considered here coincides with that discussed in more detail in section 5.2 (see Figures 5-9, F=250kN). We begin

415 by simulating the behaviour of these four structures under a flap-wise tip force of 250kN. Then, we analyze the results obtained  to show the effect of the geometric differences on their mechanical behaviour.  The results obtained are summarized in the following. In particular, Figure 10 shows the reference and deformed states of the prismatic structure (left) and the deformed states of the non-prismatic ones (right), while Figure 11 shows the displacements of their centre-line points. The main effect of the considered tip force is a displacement in the

420 z-direction in all  cases, with a displacement in the y-direction  for the  "tapered-twisted" and "tapered-twisted-curved," cases only, as it would be expected.

[Figure]

**Figure 10: Prismatic case before and after deformation (left) and non-prismatic cases after deformation (right)**

425

[Figure]

**Figure 11: Centre-line points displacement of prismatic case (blue) and non-prismatic cases (other colours) for F=250kN**

430    Similar results have also been obtained  for larger values of tip-force, F, which lead to larger tip-displacements. In particular,  hereafter we show the results for F varying from 250kN to 750kN. As for the previous test cases, the results obtained  have been compared with those from 3D-FEM simulations, confirming the computational efficiency and accuracy revealed in the previous sections. A  summary is shown in Figure 12, which provides a comparison in terms of tip-displacements, for the four geometries considered here, for F=250kN,

435    F=500kN and F=750kN. Such loads correspond, respectively, to tip-displacements of about 6.4%, 12.5% and 18.1% of the span-wise reference length. The difference between the 3D-BLM and the non-linear 3D-FEM in terms of tip-displacements is  below 0.9% in all the considered cases (Figure 12).

[Figure]

 **Figure 12: Tip displacements with 3D-BLM (blue) and 3D-FEM (red) for different geometries and  increasing F (see arrows)**

We conclude now by examining the results for the 3D strain measure $E_{11}$, also referred to as longitudinal strain, which is another important parameter for the analysis and design of rotor blades (see, for example, Griffith 2011). In particular, we focus  on the effects of taper by considering a beam-like structure with

445 bi-tapered cross-sections (the above "test case 2"). Then, we compare the outcomes of the 3D-BLM with those of linear and nonlinear 3D-FEM simulations. A  summary of the results is reported in Figure 13, which shows the maximum longitudinal strain at different reference cross-sections ( at 30%, 50%, and 70% of the span-wise reference length) and for three different tip- forces (F=250kN, F=500kN and F=750N).

[Figure]

450

**Figure 13: Max longitudinal strain $E_{11}$ in  cross-sections at 30%, 50% and 70% span for  increasing F (bi-tapered case)**

As verified by many simulations and shown in the examples, the proposed approach performs well in terms of computational efficiency and accuracy. It can be used to study the mechanical behaviour of beam-like structures, which are curved, twisted

455 and tapered in their  unstressed reference state and undergo large global displacements. It can moreover provide

information on the deformed configurations of the structures, such as their displacement fields, as well as the corresponding strain and stress measures. It is worth noting that it is suitable for beam-like structures with generic reference cross-sections shapes. However, as already pointed out, for bi-tapered elliptical cross-sections  analytical solutions can be obtained in terms of warping fields, while for generic reference cross-sections shapes  problem (29) has to be solved  using numerical methods.

**6 Conclusions**

 Many complex engineering structures,  (such as the rotor blades of wind turbines and helicopters, are non-prismatic beam-like structures, with one dimension much larger than the other two and a shape that is curved, twisted and also tapered  in the  unstressed reference state.  The increasing size and flexibility of such structures make the prediction of their aero-elastic behaviour ever more challenging. This paper addressed the structural part of this modelling and proposed a modelling approach, referred to as 3D-BLM, which is computationally efficient, accurate and explicitly consider the main geometrical characteristics of  the mentioned structures, the large deflections of their reference centre-line and the in- and out-of-plane warping of their transversal cross-sections. In the mentioned approach, the warping displacements have been  thought of as an additional small motion superimposed on the global motion of the local frames. The  strain tensor has been calculated analytically in terms of  geometrical parameters of the structure, 1D strain measures and  3D warping fields. A method based on  a variational statement has been used to obtain suitable warping fields. The  proposed approach enables to obtain analytical results  in particular cases and can be implemented into an efficient numerical code in the general case. The analytical results obtained, along with numerical examples (obtained by implementing 3D-BLM into a computer code) and comparisons with corresponding results from 3D-FEM simulations have been presented to show the effectiveness of the  modelling approach and the information it can provide. In all cases, the simulation times with 3D-BLM have been significantly shorter than those required by 3D-FEM simulations, while the accuracy of the results has been always almost the same. In this paper the analyses have been limited to the terms of order $\varepsilon$, as discussed in the introduction of the strain measures of the model. This turned out to be sufficient to accurately predict the global deflection of the considered structures even when the displacements of the centre-line points are large and nonlinear with respect to the applied loads. The inclusion of higher order terms in the model may provide better results,

especially in terms of stress and strain fields predictions, while not practically affecting the computational performance of the implemented model. This is an important point to be further investigated and will be the objective of a successive work.

490 An interesting future activity would also be to performing comparison analyses with other structural modelling methods (not only 3D-FEM), with the aim of assessing the performance of different structural models for non-prismatic beam-like structures in terms of information each approach can provide (e.g. centre-line displacements, 1D strain measures, 3D strain fields), computational efficiency and results accuracy.

**References**

[revised manuscript text omitted]